# Leveraging 3D Reconstruction for Mechanical Search on Cluttered Shelves

**Seungyeon Kim**[*,1]    **Young Hun Kim**[*,1]    **Yonghyeon Lee**[2]    **Frank Chongwoo Park**[1]

[1]Seoul National University, [2]Korea Institute For Advanced Study

{ksy, yhun}@robotics.snu.ac.kr, ylee@kias.re.kr, fcp@snu.ac.kr

**Abstract:** Finding and grasping a target object on a cluttered shelf, especially when the target is occluded by other unknown objects and initially invisible, remains a significant challenge in robotic manipulation. While there have been advances in finding the target object by rearranging surrounding objects using specialized tools, developing algorithms that work with standard robot grippers remains an unresolved issue. In this paper, we introduce a novel framework for finding and grasping the target object using a standard gripper, employing pushing and pick-and-place actions. To achieve this, we introduce two indicator functions: (i) an existence function, determining the potential presence of the target, and (ii) a graspability function, assessing the feasibility of grasping the identified target. We then formulate a model-based optimal control problem. The core component of our approach involves leveraging a 3D recognition model, enabling efficient estimation of the proposed indicator functions and their associated dynamics models. Our method succeeds in finding and grasping the target object using a standard robot gripper in both simulations and real-world settings. In particular, we demonstrate the adaptability and robustness of our method in the presence of noise in real-world vision sensor data. The code for our framework is available at https://github.com/seungyeon-k/Search-for-Grasp-public.

**Keywords:** Mechanical search, Object rearrangement, Prehensile and Non-prehensile manipulation

## 1 Introduction

Finding and grasping a desired target object on a cluttered shelf – where the target is occluded by unknown objects and initially not visible to a vision sensor – is a significant challenge in robotic manipulation. This task is further complicated when the pose of the vision sensor is fixed. In such scenarios, the robot must rearrange surrounding objects to identify the target's pose and grasp it, all while avoiding collisions with the shelf and nearby objects. The geometric characteristics of the shelf, which allow visual observations solely from the front and limit the manipulator's workspace, add another layer of complexity.

Previous research on finding and grasping the target object, termed *mechanical search*, has focused on (i) simple environments, such as flat tables or bins [1, 2, 3, 4, 5, 6, 7, 8], or (ii) complex shelf environments but with the assumption that all object shapes and poses are known [9, 10, 11, 12, 13, 14, 15]. Recent studies have ventured into more realistic shelf scenarios where the surrounding objects are unknown. However, these often depend on specialized, elongated tools to manipulate the objects within the tight shelf spaces [16, 17, 18]. The challenge of finding and grasping the target object using a standard robot gripper, devoid of specialized tools, remains unsolved. To our knowledge, we are the first to offer a practical method for mechanical search on a cluttered shelf using a standard robot gripper as shown in Figure 1.

---

[*]Equal contribution

7th Conference on Robot Learning (CoRL 2023), Atlanta, USA.

Using a standard gripper introduces new practical challenges that were not addressed by existing methods. In a shelf space densely populated with unknown objects, identifying collision-free spaces and generating collision-free actions are crucial. However, when using a standard gripper, as opposed to specialized elongated tools, accounting for collisions between the gripper, robot links, objects, and shelves becomes much more challenging (e.g., imagine scenarios in which the robot is required to reach its arm deeper into the shelf). A distinguishing feature of our approach, which serves as the solution to the aforementioned challenge and sets it apart from existing methods, is the utilization of a *3D reconstruction model*.

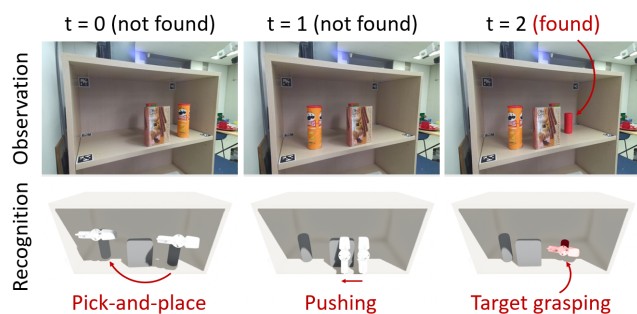

Figure 1: A 3D recognition-based mechanical search and grasping of the target object (red cylinder).

In this paper, we propose a novel optimal control framework for mechanical search with practical algorithms that leverage 3D reconstruction models. First, we introduce two indicator functions (denoting the target's candidate pose by $x \in \mathrm{SE}(3)$): (i) an existence function $f(x)$ that indicates if the target can be present at $x$ and (ii) a graspability function $g(x)$ that indicates if the target at $x$ is graspable. The objective then becomes to rearrange surrounding objects until only one existable and graspable pose $x^*$ remains, i.e., there exists a unique $x^* \in \mathrm{SE}(3)$ such that $f(x^*) = 1$ and $g(x^*) = 1$, which leads to a straightforward definition of a cost function using $f$ and $g$.

Second, we leverage a 3D object recognition model to effectively estimate the functions $f, g$ and their corresponding dynamics models; therefore we formulate a tractable model-based optimal control problem. Specifically, we employ a recent 3D recognition model rooted in *superquadric* primitives [19]. Notably, the superquadric representation allows for rapid collision checks, depth image rendering, and the utilization of pushing dynamics models [20], which leads to effective estimations of $f$ and $g$ and their dynamics models. To mitigate accumulated estimation errors during optimal control, we adopt the model predictive control with a short time horizon.

Through experiments in both simulations and real-world scenarios, we have validated the effectiveness of our 3D reconstruction-based approach. Our method is able to find and grasp the target object using a standard two-finger robot gripper, even in the presence of noise from vision sensor data in real-world settings.

## 2 Related Works

**Mechanical search.** Earlier studies have focused on algorithmic approaches to finding the target object by manipulating surrounding objects through pushing or pick-and-place actions [1, 2, 3]. Despite their contributions, these methods make many environmental assumptions and they are computationally intensive and slow. Subsequent efforts have leveraged the POMDP model and solvers like DESPOT [4] or POMCP [5]. More recent studies have employed deep learning-based modules, such as object segmentation and recognition networks, to enhance search effectiveness [6, 7]. Another significant work uses a 3D shape recognition to predict occluded geometries from the vision sensor, leveraging this information to enhance the search process [8]. Our work aligns with [8] in leveraging the 3D recognition model to effectively solve the mechanical search problem. However, we employ an implicit object representation, offering various computational advantages for the mechanical search problem.

**Mechanical search on shelves.** Given that shelves are commonly used to store objects in home environments or logistic warehouses, mechanical search on shelves has emerged as a critical research area [16, 17, 18]. The inherent constraints of shelves introduce challenges, including potential collisions with the shelf, constraints on object removal, and limited camera views. An extension of

the X-ray method [7] has been proposed to address these challenges [16]. Subsequent studies have introduced novel tools to augment the capabilities of robotic actions, such as stacking [17, 18]. Our approach is distinct because it incorporates the graspability of the target object during mechanical searches. While previous studies often utilize specialized grippers, thereby eliminating the need to consider the graspability of the target, our use of standard grippers necessitates a careful consideration of potential collisions during the grasping of the target object. This requires 3D scene reasoning, leading us to adopt a 3D recognition model, which is a departure from the 2D reasoning commonly employed in prior works. More detailed reviews on related works can be found in Appendix A.

## 3 A General Framework for Mechanical Search and Grasping

In this section, we propose a general framework for mechanical search and provide two specific algorithms: (i) search-and-grasp and (ii) search-for-grasp, using the model-based optimal control formulation. The search-and-grasp method executes actions first to find the target and then executes additional actions to rearrange the objects to make the target graspable, while the search-for-grasp method integrates the search and grasp processes into a cohesive framework so that the target object's graspability is taken into account in the search phase.

Our focus lies specifically on scenarios where the target object is known – where its information is given as the color and geometry – and is fully occluded by other unknown objects. We assume that the robot can interact with objects either by pushing or pick-and-place actions to rearrange the objects. When the robot performs the pick-and-place action, it is only allowed to place an object in another empty space on the shelf (i.e., object stacking is not allowed). For the target object, we assume that it is placed in a straight-up pose (i.e., not tilted) and not stacked on top of the other surrounding objects. Also, when the robot rearranges the surrounding non-target objects, the target object is assumed to remain stationary.

We denote the target object's pose by $x \in \mathrm{SE}(3)$; since the target object is not tilted and stacked, it is sufficient to represent the object pose as $x \in \mathrm{SE}(2)$. Instead of dealing with the continuous pose space $\mathrm{SE}(2)$, to simplify numerical computations, we restrict our attention to a finite subset $\mathcal{X} \subset \mathrm{SE}(2)$. The two core components in our framework are (i) an existence function $f : \mathcal{X} \to \{0, 1\}$ and (ii) a graspability function $g : \mathcal{X} \to \{0, 1\}$. The existence function $f(x)$ indicates whether the target object can be present at the pose $x$ or not. For example, given an observation shown in Figure 2 (*Upper*), consider two candidate target poses $x \in \mathcal{X}$ (i.e., the red cylinders in Figure 2 *Lower*). For $f(x)$ to be 1, the rendered image must match the observation (there are other conditions as well, details are in the next section), otherwise, $f(x) = 0$. In practice, considering the discretization resolution of $\mathcal{X}$, $f(x)$ is considered to be 1 if the target object pose can exist near $x$. Naturally, it captures the uncertainty of the target object's

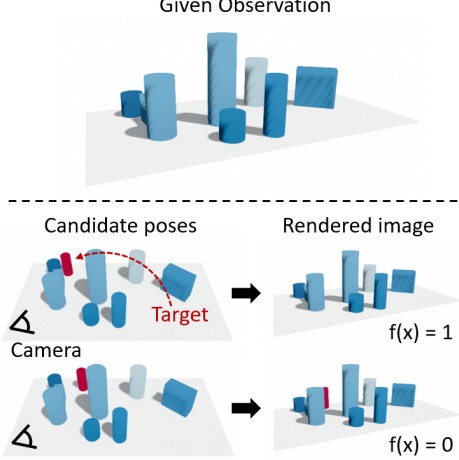

Figure 2: Illustration on the existence function $f(x)$. *Upper*: Observation. *Lower*: Candidate poses and hypothetical rendering results.

pose because $\sum_{x \in \mathcal{X}} f(x)$ represents the number of possible object poses; the greater $\sum_{x \in \mathcal{X}} f(x)$, the more uncertain. We assume $\mathcal{X}$ is sufficiently densely discretized so that $\sum_{x \in \mathcal{X}} f(x)$ is lower bounded by 1 (i.e., the target object must exist in at least one $x \in \mathcal{X}$). The graspability function $g(x)$ indicates whether the target object at the pose $x$ is graspable or not.

Denote the existence and graspability functions at a discrete timestep $t$ by $f_t(x)$ and $g_t(x)$ and a pushing or pick-and-place action by $a_t$. We assume that the dynamics models for $f_t(x)$ and $g_t(x)$ are given and denoted by $\mathcal{F}$ and $\mathcal{G}$ such that $f_{t+1}(x) := \mathcal{F}(f_t, a_t)(x)$ and $g_{t+1}(x) := \mathcal{G}(g_t, a_t)(x)$.

Additionally, we assume that the uncertainty of the target object pose is not increasing in $t$, i.e., $\sum_{x\in\mathcal{X}} f_t(x) \geq \sum_{x\in\mathcal{X}} \mathcal{F}(f_t, a_t)(x)$, regardless of $a_t$. This non-increasing uncertainty assumption comes from the prior assumption that the target object does not move when an action is applied because it is impossible for an object suddenly can exist where it could not originally exist.

Given initial $f_0$ and $g_0$ and dynamics models $\mathcal{F}$ and $\mathcal{G}$, we formulate a *search-for-grasp method* as a model-based optimal control. Specifically, the goal is to find a sequence of actions $a_0, \ldots, a_T$ for a fixed time $T$, so that $f_T(x^*) = 1$ for only one $x^* \in \mathcal{X}$ – if there are more than one, then we cannot specify at which $x$ the target object exists – and that is graspable, i.e., $g_T(x^*) = 1$. For this purpose, we formulate the following model-based optimal control problem:

$$\min_{a_1,\ldots,a_T} \sum_{x\in\mathcal{X}} f_T(x) + \alpha f_T(x)(1 - g_T(x)), \tag{1}$$

such that $f_{t+1}(x) = \mathcal{F}(f_t, a_t)(x)$ and $g_{t+1}(x) = \mathcal{G}(g_t, a_t)(x)$, and $\alpha$ is a hyperparameter. Minimizing the first term in (1) makes $f_t$ have only one $x^*$ such that $f_T(x^*) = 1$ – since the existence function is lower bounded by 1 –, and minimizing the second term enforces $g_T(x^*) = 1$. Without the second term, even though the target object becomes visible, there is no guarantee that it will be graspable.

In practice, we prefer to perform a minimum number of actions needed for the task, rather than for a pre-determined fixed length $T$. We introduce a discount factor $\gamma \in (0, 1]$ and propose a modified version of the optimal control problem:

$$\min_{a_1,\ldots,a_T} \sum_{t=0}^{T-1} \gamma^t (\mathcal{J}_{t+1} - \mathcal{J}_t) = \gamma^T \mathcal{J}_T + \frac{1-\gamma}{\gamma} \sum_{t=1}^{T} \gamma^t \mathcal{J}_t - \mathcal{J}_0, \tag{2}$$

where $\mathcal{J}_t := \sum_{x\in\mathcal{X}} f_t(x) + f_t(x)(1 - g_t(x))$. As $\gamma \to 1$, the new objective function converges to the original one in (1) since $\mathcal{J}_0$ is constant. For $\gamma < 1$, $\mathcal{J}_t$ for $t < T$ are now taken into account, and thus minimizing (2) leads to accomplishing the task quickly. A $\gamma$ that is too small can make the problem difficult, so an appropriate level of $\gamma$ needs to be found.

Additionally, we propose a computationally lighter version, a *search-and-grasp method*. In the cost term $\mathcal{J}_t$ in (2), the computation cost of $g_t(x)$ is very high since it requires the planning of multiple collision-free paths (as we explain in the next section in more details). The function $g_t(x)$ must be calculated for all $x$ such that $f_t(x) = 1$, thus as $\sum_x f_t(x)$ is bigger, the computation cost increases. In the search-and-grasp method, we first find the target by using the first term only $\mathcal{J}_t = \sum_x f_t(x)$. Then, after the target is found at $x^*$, we use the second term $\mathcal{J}_t = 1 - g_t(x^*)$ to rearrange the objects to make the target graspable. Since the target pose is already specified at the search phase, in the second stage, we need to compute the graspability function $g_t$ only at one $x^*$, which significantly reduces the total computation time.

## 4   3D Object Recognition-based Mechanical Search

In this section, as one practical way to implement our framework, we propose an object recognition-based approach. We assume that a scene contains multiple unknown objects and a known target object, and the RGB-D camera's pose is fixed so can only capture one side of the scene (e.g., a shelf containing objects from the front view). From the partial view information of the scene, our strategy is to first recognize the shapes and poses of the objects and then to use the recognition results to estimate the existence and graspability functions $f$ and $g$ and the dynamics models $\mathcal{F}$ and $\mathcal{G}$.

### 4.1   Object Recognition via Superquadrics

In this paper, we employ the shape class called the superquadrics, a family of geometric shapes which represent diverse shapes ranging from boxes, cylinders, and ellipsoids to other symmetric complex shapes. The implicit function of a superquadric surface at the canonical pose (i.e., $T =$

$I_{4\times4}$ is the identity matrix) has the following form:

$$\left(\left|\frac{x}{a_1}\right|^{\frac{2}{e_2}} + \left|\frac{y}{a_2}\right|^{\frac{2}{e_2}}\right)^{\frac{e_2}{e_1}} + \left|\frac{z}{a_3}\right|^{\frac{2}{e_1}} - 1 = 0, \tag{3}$$

where $q = (a_1, a_2, a_3, e_1, e_2) \in \mathbb{R}^5$ is the shape parameter.

Especially, $a_1, a_2, a_3$ controls the sizes and $e_1, e_2$ controls the geometric shapes (some examples are shown in Figure 3). Given a pose of the object $T \in \mathrm{SE}(3)$, the implicit surface equation is transformed accordingly. Recently, given a partial observation of an object, a model that predicts the 3d full shape of the object by using the superquadric representation has been proposed [19]; we use this model for 3D object recognition. We note that to efficiently represent more complex 3D shapes, we can use latent variable models, e.g., using the autoencoder framework [21, 22, 23, 24, 25, 26]. We include details about the model architectures and the training details in Appendix B.1.

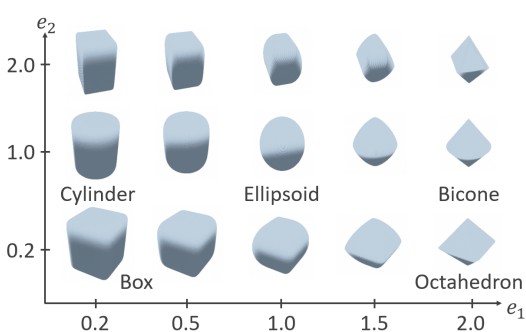

Figure 3: Examples of superquadrics.

## 4.2   Existence and Graspability Function Estimates $\hat{f}$ and $\hat{g}$

Given a partial visual observation $o$, let a set of estimated superquadric shape parameters and poses be denoted by $s = \{(q^i, T^i)\}$. Further, we assume that an indicator variable $c \in \{0, 1\}$ – indicates whether the target object is visible (i.e., $c = 1$) or not (i.e., $c = 0$) in the observation $o$ – is available. If target is visible (i.e., $c = 1$), we assume that the target object's pose can be estimated accurately. Using these estimates $s$ and $c$, we obtain the estimates for the existence and graspability functions denoted by $\hat{f}(x; s, c)$ and $\hat{g}(x; s)$.

The key idea is to locate the recognized superquadric objects in the simulation, as well as the target object at a hypothetical pose $x \in \mathcal{X}$. If $c = 1$, the existence function $\hat{f}$, since we know at which $x^*$ the target object exist, is 1 only at $x^*$, i.e., $\hat{f}(x^*; s, c = 1) = 1$ and 0 at other $x \in \mathcal{X}$. If $c = 0$, $\hat{f}(x; s, c = 0)$ is defined to be 1 if (i) depth rendering results with and without the target object at $x$ are identical and (ii) there is no collision between the recognized objects, the environment, and the target object. Otherwise $\hat{f}(x; s) = 0$. The graspability function $\hat{g}(x; s)$ is defined to be 1 if we can find a collision-free grasping trajectory of the robot gripper, where all possible collisions between the robot arm, gripper, environment, and multiple objects should be taken into account. The depth rendering, collision checking, and grasp planning can be performed efficiently using 3D recognition results, and the details on the computation of the existence and graspability functions are given in Appendix B.2 and B.3, respectively.

## 4.3   Approximate Dynamics Models $\hat{\mathcal{F}}$ and $\hat{\mathcal{G}}$

Then, we construct the approximate dynamics models for $f_t$ and $g_t$ by using (i) the dynamics of $s_t$ denoted by $s_{t+1} = \mathcal{S}(s_t, a_t)$ – where we just transform the selected object for the pick-and-place action and use pre-trained SE(2)-equivariant pushing dynamics model for pushing action [20] – and (ii) the function estimates $\hat{f}$ and $\hat{g}$. Since $\hat{f}$ takes the classification label $c$ as an input, we need a dynamics model for $c_t$, but it is hardly possible to know if the target will be visible or not in the future given an action $a_t$. We take a conservative strategy and assume that the visibility of the target object $\tilde{c}_{t+1}$ does not change at $t + 1$, i.e., $\tilde{c}_{t+1} = c_t$. Then, for given $f_t$ and $g_t$ at time $t$, the approximate dynamics models are defined as follows:

$$\hat{\mathcal{F}}(f_t, a_t)(x) := f_t(x)\hat{f}(x; s_{t+1}, \tilde{c}_{t+1}); \quad \hat{\mathcal{G}}(g_t, a_t)(x) := \hat{g}(x; s_{t+1}), \tag{4}$$

where $s_{t+1} = \mathcal{S}(s_t, a_t)$ and $\tilde{c}_{t+1} = c_t$. The existence function $\hat{\mathcal{F}}(f_t, a_t)(x)$ is 1 if both $f_t(x)$ and $\hat{f}(x; s_{t+1}, c_{t+1})$ are 1, which guarantees the non-increasing uncertainty condition for $\mathcal{F}$, i.e., $\sum_{x \in \mathcal{X}} f_t(x) \geq \sum_{x \in \mathcal{X}} \hat{\mathcal{F}}(f_t, a_t)(x)$.

## 4.4 Sampling-based Model Predictive Control

Given an initial observation $o_0$ and function estimates $f_0(x) = \hat{f}(x; s_0, c_0), g_0(x) = \hat{g}(x; s_0)$, by using the approximate dynamics models $\hat{\mathcal{F}}, \hat{\mathcal{G}}$, we can solve the optimal control problem in (2). Since both the function estimates and dynamics models have numerical errors, we perform the model predictive control (MPC) where we iterate the following procedure for $T$ times: for $t = 0, \ldots, T-1$ (i) update initial estimates of $f_t, g_t$ from a new observation $o_t$, (ii) solve (2) with a short time horizon $M < T$, and (iii) take only the first action $a_t$. Details for sampling-based model predictive control can be found in Appendix B.4.

# 5 Experiments

In this section, we conduct a comparative analysis of the two proposed methods, the search-and-grasp and search-for-grasp methods, and evaluate both methods to show robust performance to find and grasp the occluded target object in both simulation and real-world environments.

**Environment.** The simulation and real-world experiments share the same robot, the same camera, and the same shelf environment; the simulation experiments are conducted by the Pybullet simulator. We use the 7-dof Franka Emika Panda robot with a parallel-jaw gripper and an Azure Kinect DK camera sensor looking into the shelf at a fixed location (See Figure 4). The raw input visual observation from the camera is an RGB-D, which is then converted to point cloud data. In the simulation environment, we assume that the ground-truth segmentation masks of objects are available; for real-world experiments, we adopt color-based heuristic segmentation method. We use cylinder-shaped and cube-shaped objects of various sizes which are visualized in Figure 4.

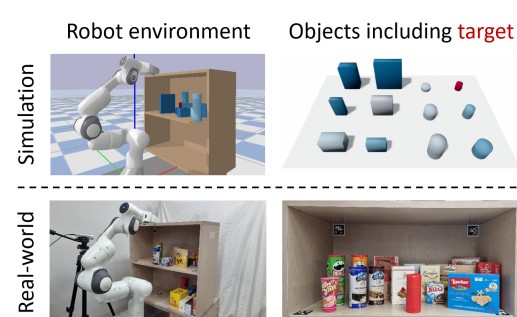

Figure 4: The left column shows the simulation and real environments, and in the right column, objects used in each environment are visualized. In particular, the target object is marked in red in the simulation; the red-taped can is the target object in the real experiment.

More details on the environment are in Appendix C.1. Throughout the experiments, we restrict our attention to a cylindrical target object. Considering the rotational symmetry of the target object and the assumption that the target object stands upright, the target pose space $\mathcal{X}$ can be reduced to $\mathbb{R}^2$. We note that if the target object is a box, the whole $\mathrm{SE}(2)$ space should be considered as the target pose space; the experimental results when the target object is a box can be found in Appendix D.3.

**Action space.** The action space $\mathcal{A}$ is composed of either pushing or pick-and-place actions. A pushing action is defined by $(T_{\mathrm{push}}, d) \in \mathrm{SE}(3) \times \mathbb{R}^3$, where $T_{\mathrm{push}} \in \mathrm{SE}(3)$ is an initial gripper pose and $d \in \mathbb{R}^3$ is a displacement of the gripper with a fixed orientation. A pick-and-place action is defined by $(T_{\mathrm{grasp}}, T_{\mathrm{place}}) \in \mathrm{SE}(3) \times \mathrm{SE}(3)$, which are gripper poses when it grasps and places the object, respectively. If the gripper collides with other objects, it is judged to be invalid and is not sampled. For details on the actions and the action sampling method, we refer to Appendix B.5.

**Evaluation metrics.** We report the find- and grasp-success rates separately. If a sequence of our actions makes the target visible within 10-time steps, it is considered as a find-success. If it makes the target not only visible but also graspable and we can find a collision-free path for taking out the object within 10-time steps, it is considered a grasp success. Additionally, we measure the average number of actions required to find or grasp the target.

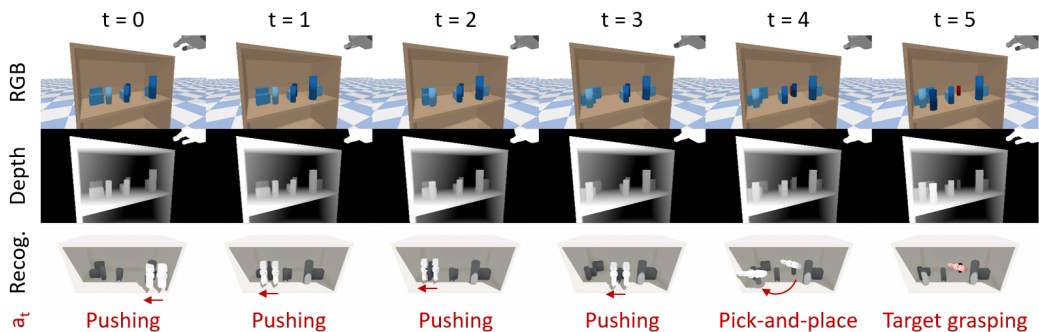

Figure 5: An example trajectory of simulation manipulation. Each column shows the camera input and action selection at each time step. In the simulation, surrounding objects are blue and the target object is red.

| METHOD | | The number of objects | | | | | | | |
|---|---|---|---|---|---|---|---|---|---|
| | | 2 | | 4 | | 6 | | 8 | |
| | | Find | Grasp | Find | Grasp | Find | Grasp | Find | Grasp |
| O-Search-and-Grasp | Succ. | 0.978 | **0.939** | **0.939** | 0.761 | 0.878 | 0.622 | 0.844 | 0.528 |
| | Steps | 1.392 | 1.562 | 2.178 | 2.635 | 2.589 | 3.473 | 2.664 | 3.947 |
| O-Search-for-Grasp | Succ. | **0.983** | 0.928 | 0.933 | **0.794** | **0.9** | **0.678** | **0.889** | **0.578** |
| | Steps | 1.407 | 1.647 | 2.077 | 2.769 | 2.377 | 3.574 | 2.831 | 4.125 |
| R-Search-and-Grasp | Succ. | **0.983** | **0.928** | 0.928 | **0.789** | 0.889 | 0.656 | 0.878 | 0.606 |
| | Steps | 1.362 | 1.611 | 2.222 | 2.739 | 2.581 | 3.432 | 2.981 | 3.78 |
| R-Search-for-Grasp | Succ. | 0.972 | 0.922 | **0.933** | 0.756 | **0.922** | **0.672** | **0.894** | **0.656** |
| | Steps | 1.331 | 1.651 | 2.196 | 2.765 | 2.301 | 3.281 | 2.901 | 3.975 |

Table 1: Simulation manipulation results

## 5.1 Simulation Experiments Results

In this section, we evaluate our methods in the simulation experiments and empirically show that they can find and grasp the fully-occluded target object. To evaluate our method, we have created 180 scenarios for each number of surrounding objects in $\{2, 4, 6, 8\}$, so a total of 720 scenarios; for each scenario, a random selection (allowing duplicates) is made among the given objects in Figure 4. More details on simulation experiment settings are provided in Appendix C.2. We note that there are some scenarios where the target object is not graspable even with the maximum number of actions, so the maximum average success rate is slightly lower than 1.

To evaluate our 3D reconstruction-based mechanical search method, and in particular to see how much recognition error affects the task performance, we also test the cases where the ground-truth poses and shape parameters of the surrounding objects (not the target object) are available, and denote them as *oracle*. In these cases, the the recognition module is not used and the ground-truth information of the surrounding objects are directly used for solving optimal control. We put the letter 'O' in front of the method name in the oracle experiments, and 'R' in the experiments using the 3D recognition (e.g., O-Search-and-Grasp and R-Search-for-Grasp).

Figure 5 shows an example of how our recognition-based search-for-grasp method acts on the simulation experiment. The search-for-grasp method succeeds in finding the target object in four pushing actions and then makes the target object graspable by performing an additional pick-and-place action. The success rates and the average number of actions for finding and grasping the target are shown in Table 1. First, the performance differences between *oracle* and *recognition* are not significant, which means that the 3D recognition error does not significantly affect performance. Second, search-for-grasp has no difference in performance from search-and-grasp when the number of surrounding objects is small, but shows better performance when the number is large. This suggests that considering graspability is of great help when finding the target. Further experimental results with more diverse scenarios are provided in Appendix D.1 and D.2.

## 5.2 Real-world Experiments Results

We adopt the R-Search-for-Grasp method for finding and grasping the desired target object in real-world environment. Figure 6 shows a real-world manipulation result. The target object is occluded by the two objects, and the target object is found through three pushing actions. The found target object is not graspable at $t = 3$, an additional pushing action is applied to make the target object graspable[1].

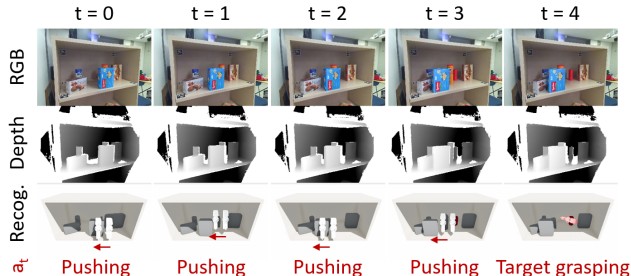

Figure 6: Search-for-grasp real-world manipulation results

**Failure cases.** Table 2 shows the manipulation success rates in real-world experiments. We design 5 test scenarios for each number of objects, and used object configurations for each experiment are in Appendix C.3. A few failure failure cases occur, especially when the number of the surrounding objects increases. Most of the reasons for failure cases are (i) that there's no solution of rearranging the objects in our designed action space (see Appendix B.5) and (ii) that incorrect 3D recognition causes erroneous updates of existence and graspability maps; for example, if the recognition model recognizes an object as inaccurately large, the existence map may be underestimated, i.e., it is decided that $f(x) = 0$, but the target object can exist at $x \in \mathrm{SE}(2)$.

| Num. | Find | Grasp |
|------|------|-------|
| 3 | 5/5 | 5/5 |
| 4 | 5/5 | 4/5 |
| 5 | 4/5 | 2/5 |
| 6 | 3/5 | 2/5 |

Table 2: Search-for-grasp real-world manipulation results.

## 5.3 Limitations and Future Directions

First, since our method considers single superquadric-shaped objects, it is not trivial to apply it directly to more complex objects. One future direction to extend our method is to exploit the researches that attempt to represent the complex objects as multiple superquadrics [27, 19, 28, 29]. Even if each object is represented with several superquadrics, we can compute the existence function and the graspability function in the same manner. Second, our method highly depends on the 3D recognition model, inaccurate recognition can cause several problems. There can be robot-object collisions because we check the collision between the robot action and recognized objects (see Appendix B). Moreover, some inappropriate action decisions can be made due to erroneous existence map estimation. To overcome the limitation, additional information such as RGB images should be utilized to enhance the recognition performance [30, 31]. Lastly, since the graspability of the object is checked using only a limited number of grasping trajectories (see Appendix B.3), there can be cases where it is determined that the object cannot be grasped even if there is a grasping trajectory. As a research direction to overcome this, the graspability can be checked with more diverse trajectories quickly using a network-based planner [32].

## 6 Conclusion

We have introduced a novel mechanical search and grasping framework for cluttered shelves and developed practical algorithms named search-and-grasp and search-for-grasp, utilizing a pre-trained 3D object recognition model. Our approach uses pushing and pick-and-place actions to rearrange occluding objects, making the desired target object both visible and graspable. Unlike prior research which often relies on specialized elongated tools, we employ a standard two-finger gripper with robot actions including pushing and pick-and-place, and 6-DoF grasping. Our method achieves a 90-65% success rate in simulation experiments depending on the number of objects varying from 2 to 8, and it maintains comparable efficacy in real-world settings, even in the observation noise.

---

[1]The real-world manipulation videos can be found at this Youtube link.

**Acknowledgments**

S. Kim, Y. H. Kim, and F. C. Park were supported in part by SRRC NRF grant RS-2023-00208052, IITP-MSIT grant 2021-0-02068 (SNU AI Innovation Hub), IITP-MSIT grant 2022-0-00480 (Training and Inference Methods for Goal-Oriented AI Agents), KIAT grant P0020536 (HRD Program for Industrial Innovation), ATC+ MOTIE Technology Innovation Program grant 20008547, SNU-AIIS, SNU-IAMD, SNU BK21+ Program in Mechanical Engineering, and SNU Institute for Engineering Research. Y. Lee was the beneficiary of an individual grant from CAINS supported by a KIAS Individual Grant (AP092701) via the Center for AI and Natural Sciences at Korea Institute for Advanced Study.

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

# Appendix

## A  Related Works

### A.1  Mechanical Search on Shelves

**Object search and mechanical search.** The goal of object search (sometimes called as target object retrieval) is to find a target object from unknown environments. Some works have focused on the *active perception* problem of making decisions of the sequence of camera poses to find a target object using a camera-mounted mobile robot [33, 34, 35, 36, 37, 38, 39]; recently, deep learning-based methods have been proposed in terms of target-driven visual navigation [40, 41, 42, 43]. However, in a more complex environment, such as a cluttered environment on a tabletop or an environment where objects are placed on a shelf, it may be impossible to find a target object by controlling only the position of the camera. To solve these issues, *interactive perception*-based methods – in which the robot can change the environment to find the target object – have been proposed. Object search using interactive perception is recently called mechanical search.

**Mechanical search methods.** The earlier works have attempted to solve the problem of searching the target object via performing pushing or grasping actions to the surrounding objects in algorithmic manners [1, 2, 3]. Although these methods have made a significant contribution to the research topic of mechanical search, many assumptions are made in the environment to make the problem tractable, and they are generally computationally complex and therefore slow. To improve these methods (e.g., relaxing the assumptions), several works have proposed a POMDP model and its solver (e.g., DESPOT [4] or POMCP [5]) for mechanical search. A recent work provides a generalized formulation of mechanical search and solve this problem effectively using deep learning-based perception module (e.g., object segmentation and recognition network) and grasping module (e.g., pre-trained Dex-Net) [6]. The follow-up paper proposes a novel perception module and a policy that minimizes the support of learned occupancy distributions obtained from the perception, and claims that the proposed method outperforms the previous methods [7]. Another work propose a 3D shape recognition-based approach that predicts the occluded geometries from the vision sensor image and then utilize this information to efficiently find the target object [8]. Our work is also in the spirit of [8] in utilizing the 3D shape recognition module to solve the mechanical search problem efficiently (e.g., reduce the number of total actions), but we use implicit representation for the recognized objects to utilize them for efficient and effective action decision (see Appendix A.3).

**Mechanical search on shelves.** As the shelves are often used to store the objects in home environments or logistic warehouses, mechanical search on shelves are being studied as an important research topic [16, 17, 18]. Object manipulation on the shelves is more challenging because of the several task constraints: the manipulator must not collide with the shelf, the objects cannot be removed from the shelf, and only a nearly-lateral camera view is available. These constraints limit the action space of the manipulator and the amount of visual information that can be obtained from the vision sensor. An earlier work proposes an extension of the previous method named lateral access X-ray [7] to solve laterally-accessible mechanical search [16]. The follow-up studies use novel tools to extend the robot action space from just pushing to pushing-and-grasping [17] and stacking [18].

The main difference from these works is that the graspability of the target object is taken into consideration when performing mechanical search. The existing studies use a custom long suction gripper specialized for mechanical search, so the graspability of the target object does not need to be considered separately. On the other hand, when using the standard robot gripper (e.g., parallel-jaw gripper), we need to find a trajectory that does not collide with the other surrounding objects and the shelf to grasp the target object. To find a non-collide trajectory, the 3D reasoning of the current scene is inevitable for 6-DOF grasping, and accordingly, a 3D recognition model has been adopted in our method. The existing works only perform 2D reasoning for the scene since they do not have to consider the graspability, and the point that we perform mechanical search task adopting 3D recognition is also different from other works.

## A.2  Object Rearrangement for Target Object Grasping

The object rearrangement generally refers to the problem of finding the feasible paths of the objects that move the objects from their initial configuration to desired final configuration, and in fact, a lot of various object rearrangement studies has been conducted; in this subsection, we only focus on the object rearrangement researches for grasping the target object. An earlier work propose an algorithm to remove the surrounding objects using prehensile manipulation to grasp the target object without robot-object collisions [9]. Since the action space is limited only by prehensile manipulation, object rearrangement algorithms using non-prehensile manipulation have also been conducted; for example, these algorithms are based on tree-search [10], persistent homology [11], and semi-autonomous tele-operation [12]. We note that unlike mechanical search, these papers assume that the information about the target object (and sometimes information about the environment) is known. Other works focus on more general cases where the target object is possibly occluded [13, 14, 15]. If the target object is occluded, the proposed performs an algorithm to find the target similar to the mechanical search. It is worthy to note that our problem is more challenging since the surrounding objects can be removed in previous studies, but cannot in our case. Also, these studies first find the target object and then grasp it when the target is occluded; we argue in this paper that finding a target object while simultaneously considering whether it can be graspable is more efficient.

**Grasping the invisible.** It is valuable to note that our problem setting is the closest to the problem considered in [44]. Their work also considers the problem of grasping the target object while considering the mechanical search problem. They named this problem *grasping the invisible* and introduce a deep learning-based end-to-end method, more specifically, a critic function that maps the visual observations to the expect rewards of robot pushing or grasping actions. This paper is the same in that it addresses the same problem as ours, but the proposed methods so far are limited to a specific environment and may require a lot of data for the model to generalize to other environments. We develop a method that can be applied in various environments by using object recognition, which is known to be well generalizable to unseen scenes [19, 20], rather than an end-to-end method.

## A.3  Shape Recognition-based Robot Manipulation

Numerous approaches have been proposed for the recognition of complete 3D shapes based on partial observations like depth images. Some of these methods employ explicit representations such as occupancy grid [45], point cloud [46], or mesh [47]. However, due to the limited resolution of these representations, they often result in imprecise shape predictions. To address this issue, recent studies have explored the use of neural implicit functions to learn implicit 3D representations for objects [48, 49, 50, 51]. In our research, we utilize superquadric functions, which strike a balance between shape expressiveness, computational efficiency, and the number of parameters required [52]. Superquadric functions have found applications in robotic manipulation tasks such as grasping [53, 54, 55]. Although we represent each object as a single superquadric function in our paper, our approach can be easily extended to encompass general implicit representations, particularly deformable superquadrics [56, 19] or a collection of superquadrics [27]. To efficiently express more complex 3D shapes, we can use latent variable models, e.g., autoencoder framework [21, 22, 23, 24, 25, 26], with sufficiently low-dimensional latent vectors.

# B Implementation Details for Our Methods

## B.1 Details for Object Shape Recognition

The object shape recognition is an algorithm that takes a partial observation from a (synthetic or real-world) RGB-D camera as input and outputs the 3D shapes of the objects in the scene. Especially, the input is a partial point cloud of the scene obtained from a depth camera $\mathcal{P} \subset \mathbb{R}^3$ (the RGB image is only used to check whether the target object is detected or not) and output is the superquadric representations $\{q_i, T_i\}_{i=1}^N$, where $q_i$ is the shape parameter, $T_i \in \mathrm{SE}(3)$ is the pose, and $N$ is the number of the objects. To design a model that performs this task, we use the same method as in the previous work [20]. We first segment a partially observed point cloud $\mathcal{P}$ into a set of object point clouds $\{\mathcal{P}_i\}_{i=1}^N$ and then convert each segmented point cloud $\mathcal{P}_i$ to superquadric representation $(q_i, T_i)$. The segmentation and superquadric recognition processes are based on neural network models. Each model is trained from synthetic dataset obtained through simulation environment, and the trained models are directly applied to both the simulation environment and the real-world environment. The overall object shape recognition process is described in Figure 7.

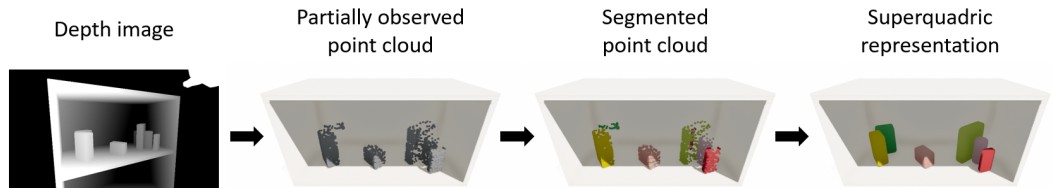

Figure 7: Overall process for object shape recognition.

**Point cloud processing.** For a partially observed point cloud $\mathcal{P}$ obtained from **simulation environment**, to bridge the sim-to-real gap, a noise is added to each point of the point cloud $\mathbf{x} \in \mathcal{P}$ using the map $\mathbf{x} \mapsto \mathbf{x} + m\mathbf{v}$ where $\mathbf{v}$ is uniformly sampled on $\mathbb{S}^2$ and $m$ is sampled from a Gaussian with zero-mean and standard deviation 0.001, as done in [57]. The all points corresponding to the shelf in the point cloud are removed and up/downsampled so that the number of points is 2048, i.e., $\mathcal{P} = \{\mathbf{x}_j \in \mathbb{R}^3\}_{j=1}^n$, where $n = 2048$. For a partially observed point cloud $\mathcal{P}$ obtained from **real-world environment**, the shelf points are removed through the known shelf shape information and pose; since there's noise on shelf pose and observed point cloud, we additionally remove the points corresponding to the floor of the shelf where the objects are placed through RANSAC plane fitting. Then as in the case of the simulation environment, the point cloud is up/downsampled so that the number of points is 2048.

**Point cloud segmentation.** After the point cloud $\mathcal{P}$ is processed, it is separated into several object point clouds $\{\mathcal{P}_i\}_{i=1}^n$ via a segmentation network. For the network architecture, we use the same architecture used in [58]. To train the segmentation network, we first find a bipartite matching between ground-truth and predicted segmentation labels using the Hungarian matching algorithm [59], and then define the loss function as the segmentation loss between the matched labels. This loss function is invariant to the point permutation of prediction, so the network can learn permutation-invariant segmentation labels and can be trained faster and more accurately accordingly.

**Superquadric recognition.** Each segmented object point cloud $\mathcal{P}_i$ is then converted to the 3D full shape represented by superquadric $q_i \in \mathbb{R}^5$ and $T_i \in \mathrm{SE}(3)$ via a recognition network proposed in [19, 20]. For the network architecture, we use the same architecture used in [20]; the input representation is a point cloud with 4-dimensional points $\mathcal{P}'_i = \{\mathbf{x}_{ij} \in \mathbb{R}^4\}_{j=1}^n$ – for each point $\mathbf{x}_{ij}$, the first three components of $\mathbf{x}_{ij}$ is equal to $\mathbf{x}_j$ and the last element of each point is 1 if $\mathbf{x}_{ij} \in \mathcal{P}_i$ and 0 otherwise for all $j = 1, ..., n$ – and output representation is $(q_i, T_i)$. The input and output of the superquadric recognition model are described in Figure 8. To train this recognition network, we adopt the training loss function as the difference between ground-truth and predicted object shapes. For the ground-truth shape, we use the point cloud uniformly sampled from the surface of the object

$\mathcal{P}_i^g = \{\mathbf{x}_j^g \in \mathbb{R}^3\}_{j=1}^{n_g}$ where $n_g = 512$. Then we use the distances from the ground-truth point cloud to the predicted superquadric as the loss function. Especially, the distance form $\delta$ proposed in [60] is used. The distance $\delta$ is defined as follows: for the superquadric surface equation $S$ expressed as

$$S(x,y,z) = \left( \left| \frac{x}{a_1} \right|^{\frac{2}{e_2}} + \left| \frac{y}{a_2} \right|^{\frac{2}{e_2}} \right)^{\frac{e_2}{e_1}} + \left| \frac{z}{a_3} \right|^{\frac{2}{e_1}}, \tag{5}$$

the distance $\delta$ between a point $\mathbf{x} \in \mathbb{R}^3$ and a superquadric surface $S(\mathbf{x}) - 1 = 0$ defined by

$$\delta(\mathbf{x}_0, S) = \|\mathbf{x}\| \left| 1 - S^{-\frac{e_1}{2}}(\mathbf{x}) \right|, \tag{6}$$

where $\|\cdot\|$ denotes the Euclidean norm. Accordingly, the training loss function for the recognition network is defined as:

$$\mathcal{L} = \frac{1}{n_g} \sum_{j=1}^{n_g} \delta^2(T^{-1}\mathbf{x}_j^g, S). \tag{7}$$

where $S$ is defined by the predicted superquadric parameters $q = \{a_1, a_2, a_3, e_1, e_2\}$ and $T$ is the predicted object pose.

We generate a training dataset to train the above two networks, the segmentation network and the superquadric recognition network. To generate data, we randomly generate $N$ objects consisting of cubes and cylinders with various shape parameters (i.e., width, height, depth for the cube, and radius, and height for the cylinder); the number of objects $N$ varies from 2 to 8. The generated objects are then placed in a random position and orientation on the shelf. After that, we construct a data tuple consisting of 3 components: (i) par-

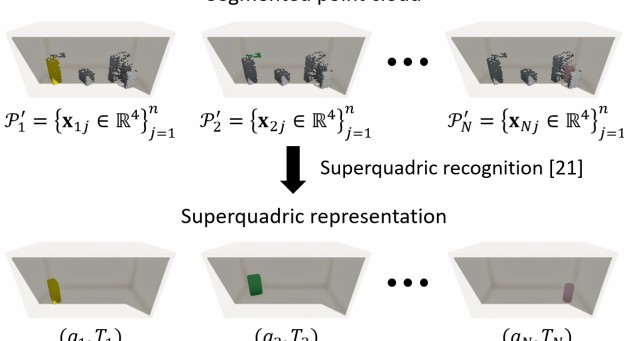

Figure 8: Input and output representation of the superquadric recognition model.

tially observed point cloud $\mathcal{P}$ from depth camera, (ii) segmentation label for each point on $\mathcal{P}$, and (iii) ground-truth point cloud (i.e., point cloud sampled from ground-truth shape) for each object $\mathcal{P}_i^g$ for $i = 1, ..., N$. For each number of objects $N$, we collect data until the numbers of the data tuples become 5000/100 for training/validation set, respectively. As a result, the total numbers of the data tuples are 35000/700 for training/validation set, respectively. The validation set is used to select the best model for the segmentation network and the superquadric recognition network.

## B.2 Details for Existence Function Estimate $\hat{f}$

In this section, we describe how to calculate the existence function estimate $\hat{f}$. The function $\hat{f}(x; s, c)$ is defined by the recognized superquadric parameters $s = \{(q_i, T_i)\}$ and the visibility of the target object $c \in \{0, 1\}$. The function's input is a hypothetical pose $x \in \mathcal{X} \subset \mathrm{SE}(3)$ and output is an indicator whether the target object can be present at the pose $x$ (i.e., $\hat{f}(x; s, c) = 1$) or not (i.e., $\hat{f}(x; s, c) = 0$). The calculation of $\hat{f}$ is trivial if $c = 1$ since the existence function $\hat{f}$ – we know at which $x^*$ the target object exist – is 1 only at $x^*$, i.e., $\hat{f}(x^*; s, c = 1) = 1$ and 0 at other $x \in \mathcal{X}$. So we consider only the case where $c = 0$, i.e., the target object is not visible. If $c = 0$, $\hat{f}(x; s, c)$ is defined to be 1 if (i) depth rendering results with and without the target object at $x \in \mathcal{X}$ are identical and (ii) there is no collision between the recognized objects, the environment, and the target object. Otherwise $\hat{f}(x; s) = 0$. We describe how to compute the above two conditions by taking advantage of the superquadric as an implicit function.

**Depth rendering condition.** We check the depth rendering condition, i.e., calculate the function $\hat{f}_d : \mathcal{X} \to \{0, 1\}$ where $\hat{f}_d(x) = 1$ if the depth rendering results with and without the target object at

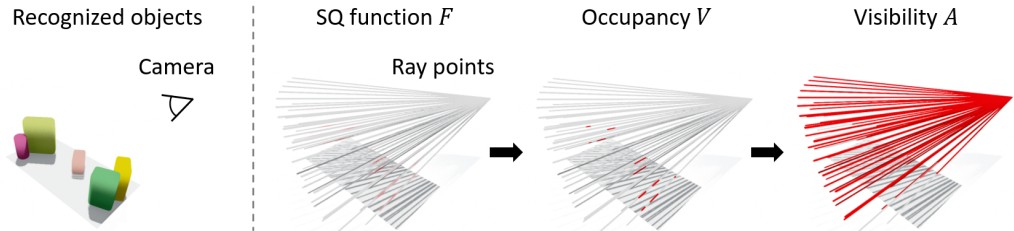

| Recognized objects | SQ function $F$ | Occupancy $V$ | Visibility $A$ |
| Camera | Ray points | | |

Figure 9: Overview of depth image rendering process from recognized superquadric functions.

$x \in \mathcal{X}$ are identical and $\hat{f}_d(x) = 0$ otherwise. To calculate this, a depth rendering function that takes the 3D object shapes $s$ as input and outputs the corresponding depth image $D \in \mathbb{R}^{H \times W}$ is required; the intrinsic and extrinsic parameters of the camera used for depth rendering are known. In this paper, depth image can be rendered quickly using the superquadric implicit function. The overview of the depth rendering process from recognized superquadric functions is described in Figure 9.

We first obtain the camera rays from the camera's intrinsic parameters, extrinsic parameters, and the resolution of the camera as described in [61], and we denote each ray's equation corresponding to the $(k, l)$-th pixel of the depth image as $r_{kl} : [t_n, t_f] \to \mathbb{R}^3$ where $k = 1, ..., H$, $l = 1, ..., W$, and $t_n$ and $t_f$ are the near and far bounds to measure distance; in this paper, we set this values as 0 and 1.5, respectively. Each ray is a straight line $r_{kl}(t) = o + td_{kl}$ where $o \in \mathbb{R}^3$ is the position of the camera pose and $d_{kl} \in S^2$ is a direction vector of the ray. The camera rays are shown in Figure 9.

After recognizing the objects, we obtain $N$ superquadric parameters and their poses $s = \{(q_i, T_i)\}_{i=1}^N$. We recall that the superquadric equation with the superquadric parameters $q_i$ is expressed by

$$S(x, y, z; q_i) = \left( \left| \frac{x}{a_1} \right|^{\frac{2}{e_2}} + \left| \frac{y}{a_2} \right|^{\frac{2}{e_2}} \right)^{\frac{e_2}{e_1}} + \left| \frac{z}{a_3} \right|^{\frac{2}{e_1}}, \tag{8}$$

where $q_i = (a_1, a_2, a_3, e_1, e_2)$. Then, we convert the parameters from recognition to implicit functions $S_i(\mathbf{x}) = S^{e_1}(T_i^{-1}\mathbf{x}; q_i) = 1$ for $i = 1, ..., N$. Using the obtained implicit functions, we calculate the signed distance-like superquadric function $F$ as:

$$F(\mathbf{x}) = \min_i S_i(\mathbf{x}) \tag{9}$$

Using this superquadric function, we can approximately calculate the occupancy function $V(\mathbf{x})$ where $V(\mathbf{x}) = 1$ if the point $\mathbf{x}$ is occupied and $V(\mathbf{x}) = 0$ otherwise. The occupancy can be calculated using the equation

$$V(\mathbf{x}) = \sigma(p(1 - F(\mathbf{x}))), \tag{10}$$

where $\sigma$ is the sigmoid function and $p$ is a scaling factor that adjusts the sharpness of $V$ – we set $p = 1000$ – as proposed in [62, 63]. Then the visibility function $A_{kl}(t)$ – which indicates whether a point is visible in the rendered camera image – on a ray point $r_{kl}(t)$ can be obtained as

$$A_{kl}(t) = \exp(-\tau \int_{t_n}^{t} V(r_{kl}(t'))dt'), \tag{11}$$

with large enough $\tau$ – we set $\tau = 100$ – as proposed in [64]. Finally, the depth value $D_{kl}$ on the $(k, l)$-th pixel of the rendered depth image $D$ is calculated as

$$D_{kl} = t_n + \int_{t_n}^{t_f} A_{kl}(t)dt, \tag{12}$$

where $k = 1, ..., H$ and $l = 1, ..., W$. The integrals above can be calculated numerically after uniformly dividing $t \in [t_n, t_f]$.

The integration is required for all $(k, l)$ pairs to obtain a perfect depth image $D$, but it is not necessary to calculate for all $(k, l)$ pairs to check the depth rendering condition. Instead, we additionally

propose an algorithm that (i) finds pixels around the recognized object (i.e., $S_i$'s) and the target object and (ii) calculates the depth value only for those pixels.

We additionally propose a modified method to calculate the depth rendering condition efficiently. The key idea is that since our region of interest is the objects on the shelf, the above calculation will also be done only near the objects. The modified algorithm is based on the fact that an arbitrary superquadric $(q, T)$ can be contained in an ellipsoid $E$; in detail, for $q = (a_1, a_2, a_3, e_1, e_2)$, the ellipsoid equation $E$ is given by

$$E(x, y, z) = \frac{x^2}{3a_1^2} + \frac{y^2}{3a_2^2} + \frac{z^2}{3a_3^2}, \tag{13}$$

and the superquadric $(q, T)$ is contained in the ellipsoid $E(T^{-1}\mathbf{x}) \leq 1$.

Using this fact, the modified method (i) reduces the number of camera rays $r_{kl}$ required for calculation, and (ii) efficiently samples the ray points $r_{kl}(t)$ required for calculation in each ray $r_{kl}$. For the first one, for each ray's equation $r_{kl}(t) = o + td_{kl}$, we calculate whether $r_{kl}(t)$ meets the ellipsoids $E(T^{-1}\mathbf{x}) \leq 1$. If $r_{kl}(t)$ does not meet any ellipsoid of the object, the depth value $D_{kl}$ is set to $t_f$, and otherwise, $D_{kl}$ is calculated following (12). Therefore, we only need to consider the rays that meet at least one of the object ellipsoids. For the second one, instead of uniformly dividing $[t_n, t_f]$ for numerical integration of (11) and (12), we use more efficient method for dividing; we obtain the intersection points between the ray and the ellipsoid, and then uniformly divide the chord corresponding to the intersections. Since all of the above calculations exist in closed-form, they have little effect on the amount of calculation.

Using the above depth rendering module, we finally obtain the function $\hat{f}_d : \mathcal{X} \to \{0, 1\}$ to check the depth rendering condition; especially, $\hat{f}_d(x) = 1$ if the MSE (Mean Squared Error) between the rendered depth images with and without the target object at $x \in \mathcal{X}$ is lower than the threshold – we set this threshold as 0.001 – and $\hat{f}_d(x) = 0$ otherwise.

**Collision condition.** We check the collision condition, i.e., calculate the function $\hat{f}_c : \mathcal{X} \to \{0, 1\}$ where $\hat{f}_c(x) = 1$ if there is no collision between the recognized objects, the environment, and the target object at $x \in \mathcal{X}$ and $\hat{f}_c(x) = 0$ otherwise. This function also can be quickly evaluated using the superquadric implicit function.

We first obtain the target object point cloud sampled from the surface of the target object $\mathcal{P}_o = \{\mathbf{x}_j^o \in \mathbb{R}^3\}_{j=1}^{n_o}$, where $n_o = 512$. Similar to the case of depth rendering condition, we also get the superquadric implicit functions $S_i$ for $i = 1, ..., N$ from the recognized objects. We note that a point $\mathbf{x} \in \mathbb{R}^3$ is inside the $i$'th object $S_i$ when the value $S_i(\mathbf{x})$ is less than 1 and otherwise outside. We utilize this fact to check whether the target object collides with the recognized objects or not. The function $\hat{f}_c : \mathcal{X} \to \{0, 1\}$ is defined as:

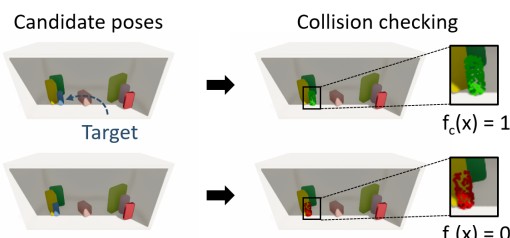

Candidate poses     Collision checking

Target

$f_c(x) = 1$

$f_c(x) = 0$

Figure 10: Illustration on the collision condition $f_c(x)$. Candidate poses and collision checking results.

$$\hat{f}_c(x) = \mathbb{1}(\min_{i,j} S_i(\mathbf{x}_j^o) > 1) \tag{14}$$

where $i = 1, ..., N$ is the object index, $j = 1, ..., n_o$ is the point index of the target object point cloud, and $\mathbb{1}(\cdot)$ is the indicator function. The calculation of the collision condition is described in Figure 10.

**Existence function estimate.** Using the above functions to check the conditions, $\hat{f}_d$ and $\hat{f}_c$, we define the existence function estimate $\hat{f} : \mathcal{X} \to \{0, 1\}$ as follows,

$$\hat{f}(x) = \hat{f}_d(x)\hat{f}_c(x).$$

In the other words, the existence function is 1 at a pose $x \in \mathcal{X}$ if both the depth rendering condition and the collision condition holds and 0 otherwise.

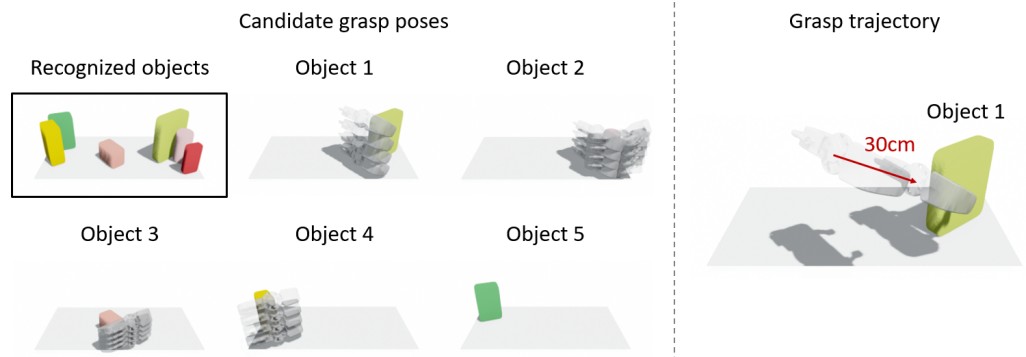

Figure 11: The examples of the candidate grasp poses for various object shapes (*Left*) and the robot trajectory for a selected grasp pose (*right*).

**Computational cost.** The calculation time takes $0.013$ seconds in average where $WH = 18,564$, $n_p = 7$, $N = 4$, and $n_o = 512$ with GeForce RTX 3090 and Gen Intel(R) Core(TM) i9-11900K @ 3.50GHz. We design the existence function is calculated in batch-wise with respect to the target pose $x$ to accelerate the computation time for MPC.

### B.3  Details for Graspability Function Estimate $\hat{g}$

In this section, we describe how to calculate the graspability function estimate $\hat{g}$. The function $\hat{g}(x; s)$ is defined by the recognized superquadric parameters $s = \{(q_i, T_i)\}$. The function's input is a hypothetical pose $x \in \mathcal{X} \subset \mathrm{SE}(3)$ and output is an indicator whether a collision-free grasping trajectory of the robot gripper – all possible collisions between the robot arm, the robot gripper, the shelf, and the surrounding objects should be taken into account – can be find at the pose $x$ (i.e., $\hat{g}(x; s) = 1$) or not (i.e., $\hat{g}(x; s) = 0$). We then describe how to compute the graspability function by also taking the advantage of the superquadrics as an implicit function similar to the calculation of the existence function.

**Candidate grasping trajectories.** To check the graspability, i.e., to check whether a collision-free grasping trajectory exists, we generate candidate grasping trajectories to grasp the target object. To achieve this, we first generate candidate grasp poses for the target object using a simple rule-based method as introduced in [20]. In this paper, we generate side grasp poses according to the shape and size of the superquadric. The pairs of the antipodal points of the target objects are sampled since we use two-finger gripper, and the

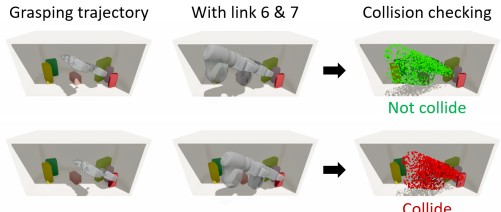

Figure 12: Illustration on the grasp trajectory collision detection. Candidate grasp trajectories and collision checking results.

grasp poses with a distance of bigger than 7.5cm between antipodal points are rejected since the maximum gripper width of the Franka gripper is 8cm. To stably grasp the target object on the shelf, we tilted the grasp poses about $20°$ relative to the ground. Accordingly, we generate $N_{gr}$ grasp poses $\{T_k^{gr}\}_{k=1}^{N_{gr}}$ where $N_{gr}$ is between 10 and 30; the examples of the grasp poses for various object shapes are shown in the left of Figure 11. Then, we additionally have to plan the trajectory of the robot arm for each grasp pose. For a grasp pose $T_k^{gr} \in \mathrm{SE}(3)$, we design a robot trajectory where the gripper approaches about 30cm along the z-direction of the gripper frame as shown in the right of Figure 11. Finally, we get $N_{gr}$ candidate grasping trajectories for the target object.

**Grasp trajectory collision detection.** After generating grasping trajectories, we should check whether the robot arm following the trajectory collides with the surrounding objects or the shelf. We first obtain the afterimage mesh of the gripper and Franka's links 7 and 6 when the robot fol-

lows the trajectory of the grasp pose $T_k^{gr}$ and obtain the point cloud $\mathcal{P}_k^{gr} = \{\mathbf{x}_{kj}^{gr} \in \mathbb{R}^3\}_{j=1}^{n_{gr}}$ sampled from the afterimage mesh, where $n_{gr} = 2048$. We get the superquadric implicit functions $S_i(\mathbf{x}) = S^{e_1}(T_i^{-1}\mathbf{x}; q_i) = 1$ for $i = 1, ..., N$ from the recognized surrounding objects. We additionally represent the shelf as the superquadric implicit functions; a shelf can be represented by five boxes so five implicit functions $S_{N+1}(\mathbf{x}), ..., S_{N+5}(\mathbf{x})$ are additionally considered. We recall from the collision condition of existence function that a point $\mathbf{x} \in \mathbb{R}^3$ is inside the $i$'th object $S_i$ when the value $S_i(\mathbf{x})$ is less than 1 and otherwise outside. The collision function $\hat{g}_c : \mathcal{X} \to \{0, 1\}$ is defined as:

$$\hat{g}_c(x) = \mathbb{1}(\max_k \min_{i,j} S_i(\mathbf{x}_{kj}^{gr}) > 1) \tag{15}$$

$$= 1 - \mathbb{1}(\max_k \min_{i,j} S_i(\mathbf{x}_{kj}^{gr}) <= 1) \tag{16}$$

where $i = 1, ..., N+5$ is the object index, $j = 1, ..., n_{gr}$ is the point index of the target object point cloud, $i = 1, ..., N_{gr}$ is the grasp pose index, and $\mathbb{1}(\cdot)$ is the indicator function. The calculation of the collision of the grasp trajectories is described in Figure 12. In practice, the modified collision detection function $\hat{g}_{c,m} : \mathcal{X} \to \{0, 1\}$ we used is:

$$\hat{g}_{c,m}(x) = 1 - \min_k \sum_i \mathbb{1}(\min_j S_i(\mathbf{x}_{kj}^{gr}) <= 1) \tag{17}$$

**Graspability function.** We define the graspability function estimate $\hat{g} : \mathcal{X} \to \{0, 1\}$ as the same with $\hat{g}_c$, i.e.,

$$\hat{g}(x) = \hat{g}_c(x). \tag{18}$$

The modified graspability function estimate is the same with $\hat{g}_{c,m}$, i.e.,

$$\hat{g}(x) = \hat{g}_{c,m}(x). \tag{19}$$

**Computational cost.** The calculation time takes 0.0128 seconds in average, where $N + 5 = 6$, $n_{gr} = 2048$, and $N_{gr} = 32$ with GeForce RTX 3090 and Gen Intel(R) Core(TM) i9-11900K @ 3.50GHz.

## B.4 Details for Sampling-based Model Predictive Control

To solve the iterative optimization in MPC (where we set $T = 10$), we use $M = 3$ and take a sampling-based approach. The actions space $\mathcal{A}$, whose elements are either pushing or pick-and-place actions, is defined based on the object recognition results and as a discrete set that can be efficiently searched through sampling (details are included in Appendix B.5). We stop the MPC iteration if the target is visible and graspable, so the total number of actions can be less than $T$. The objective function is slightly modified so that, in $\mathcal{J}_t = \sum_{x \in \mathcal{X}} f_t(x) + f_t(x)(1 - g_t(x))$, the graspability function $g_t(x)$ – which originally could take 0 or 1 – now can take $1, 0, -1, -2, -3, \ldots$ where $g(x) = -n + 1$ indicates that generated grasping trajectories for the target at $x$ collides with at least $n$ objects. This modified objective function provides more dense signals, making the optimization problem easier to solve, while not changing the optimal solution.

## B.5 Details for Action Sampling on Manipulation

In this section, we describe how actions are sampled when performing mechanical search and provide more rigorous information about actions, including pick-and-place action and pushing action. The details of the pushing action and the pick-and-place action including sampling processes and scene prediction are described in Figure 13.

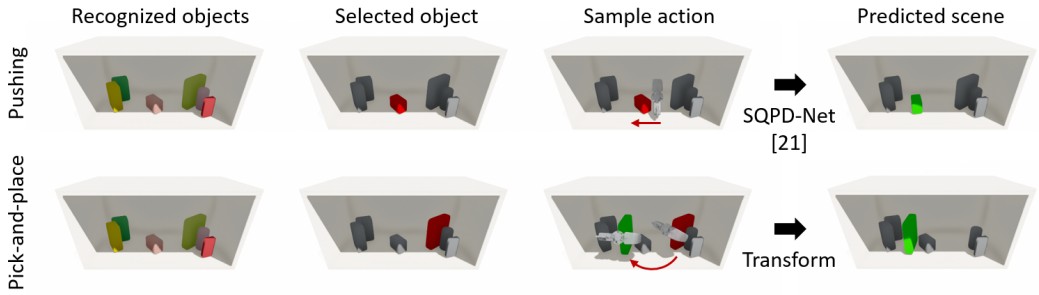

Figure 13: Sampling process and predicted scene after applying the action for pushing and pick-and-place actions.

**Pushing action.** A pushing action is defined by $(T_{\mathrm{push}}, d) \in \mathrm{SE}(3) \times \mathbb{R}^3$, where $T_{\mathrm{push}} \in \mathrm{SE}(3)$ is an initial gripper pose and $d \in \mathbb{R}^3$ is a displacement of the gripper. To sample the initial gripper pose $T_{\mathrm{push}}$, an index $i_r$ from the recognized object indices $i = 1, ..., N$ is selected and then a direction (left or right) for pushing the selected object $i_r$ is selected. Then $T_{\mathrm{push}}$ is defined from the pose and the shape parameters of the selected object $(q_{i_r}, T_{i_r})$ and the direction to push; (i) the gripper is tilted about 30 degrees along the $y$-axis of the gripper frame and (ii) the distance between the selected object and the gripper (i.e., Chamfer distance between point clouds sampled from the meshes) is set to 1cm. The pushing action is described in Figure 14. If the gripper at the pose $T_{\mathrm{push}}$ collide with the surrounding objects, the action is rejected; for colli-

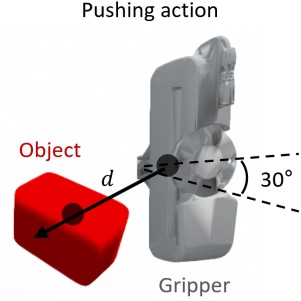

Figure 14: Visual description of the pushing action.

sion checking, we use the method used in Section B.2. The displacement of the gripper $d$ is sampled from a discrete set $\{5, 10, 15\}$cm. For each sampled action, we predict the next state $s_{t+1}$ using a pre-trained pushing dynamics model named SQPD-Net [20].

**Pick-and-place action.** A pick-and-place action is defined by $(T_{\mathrm{grasp}}, T_{\mathrm{place}}) \in \mathrm{SE}(3) \times \mathrm{SE}(3)$, which are gripper poses when it grasps and places the object, respectively. To sample the grasp pose $T_{\mathrm{grasp}}$, an index $i_g$ from the recognized object indices $i = 1, ..., N$ is selected and a grasp pose is selected from the candidate grasp poses of the selected object $(q_{i_g}, T_{i_g})$; we generate the grasp poses using the same strategy we used in Section B.3. Then a gripper pose to place $T_{\mathrm{place}}$ is selected from the poses in which the grasped object does not overlap (i.e., do not collide with) other surrounding objects and the shelf; we use the collision checking method used in Section B.2. We check the collision of the trajectory when grasping and placing and the action is rejected if there is a collision; we use the same method we used in Section B.3 when checking collision. For each sampled action, we predict the next state $s_{t+1}$ by just applying the transformation $T_{\mathrm{grasp}}^{-1} T_{\mathrm{place}}$ to the selected object.

**Action sampler.** We first check which objects can be pushed or grasped (for pick-and-place action); whether an object can be pushed or grasped is noted in the action description section above. Let $I_p = \{i_{p1}, \ldots, i_{pN_p}\}$ and $I_g = \{i_{g1}, \ldots, i_{gN_g}\}$ be the set of indices of the graspable and pushable objects, respectively. Then, we randomly choose 30 indices (allowing duplicates) from the multi-set $G + P := \{i_{g1}, \ldots, i_{gN_g}, i_{p1}, \ldots, i_{pN_p}\}$. For each chosen index, we select one action (pushing action if index is in $I_p$ and pick-and-place action if index is in $I_g$) through the method described above. The action sequences for MPC are sampled in the similar manner.

# C  Experimental Details

## C.1  Additional Details for Environment

As we note in Section 4, the simulation and real-world experiments share the same environment including the robot arm, the RGB-D camera, and the shelf. For this reason, we first build an environment in the real-world and import this environment into simulation. The intrinsic parameters of the camera and the shapes of the robot and shelf are accessible in advance; in order to match the two environments equally, the relative poses of the shelf and the camera to the robot base (i.e., relative $SE(3)$) are required. The shelf pose relative to the robot is obtained by measuring with a ruler, and the camera pose relative to the robot is obtained from the calibration using the shape of the robot arm. More specifically, the camera pose is obtained through ICP (Iterative Closest Point) algorithm between the point cloud of the robot observed by the camera and the point cloud of the robot obtained from the known shape of the robot.

## C.2  Additional Details for Simulation Experiments

**Object configuration.** We have created 180 scenarios for each number of surrounding objects in $\{2, 4, 6, 8\}$, so a total of 720 scenarios; for each scenario, a random selection (allowing duplicates) is made among the given objects.

**Initial scene setting.** We first randomly drop the selected objects on the shelf. Then, we place the (red) target object on the shelf where it is not visible from the current camera image. If there is no place to place the target object, or the target object is not graspable because of the collision of the gripper with the shelf, the scenario is discarded.

**Target detection.** In the simulation environment, a ground-truth segmentation mask can be used from the synthetic camera. If a part of the target object is observed for more than 100 pixels on the camera image (c.f. the resolution of the camera is $1280 \times 720$), it is considered to have succeeded in finding the target object in that scenario.

## C.3  Additional Details for Real-world Experiments

**Object configuration.** We have created 5 scenarios for each number of surrounding objects in $\{3, 4, 5, 6\}$, so a total of 20 scenarios; for each scenario, a pre-defined object set is used according to the number of surrounding objects as shown in Figure 15.

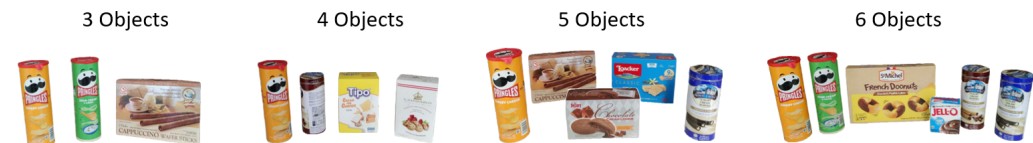

Figure 15: Pre-defined object set used for real-world experiments.

**Initial scene setting.** We put the given surrounding objects and the target object so that the target object (red cylinder) is not visible in the initial camera view.

**Target detection.** In the 3D recognition process, we segment the observed point cloud. We first calculate the average RGB value of the points in each segmented point cloud. Then, we calculate the MSE between these average RGB values and the RGB of the target object (in this case, $[0.7282, 0.1558, 0.2099]$), and if there is exactly one segmented point cloud with MSE smaller than 0.1, the target object is said to be found.

# D  Additional Experimental Results

## D.1  3-Object Toy Experiment

In this experiment, we compare the performance of the two proposed methods, search-and-grasp and search-for-grasp, and especially highlight the advantages of the search-for-grasp method. Let consider a situation where a yellow cylindrical target object is fully occluded by two larger boxes as described in the left of Figure 16. The search-and-grasp method may successfully find the target object by rearranging the two boxes, but

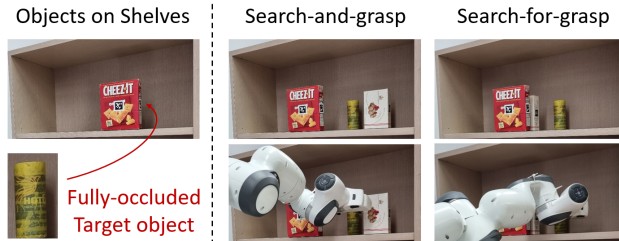

Figure 16: Comparison of *search-and-grasp* and *search-for-grasp* methods to find the target object (yellow cylinder). *This figure is a conceptual figure, not the result of implementing the methods.*

it does not guarantee the target object's graspability. As illustrated in the middle of Figure 16, the identified object may not have a collision-free robot trajectory for grasping in some cases. Consequently, additional actions would be necessary to manipulate the environment and make the target object graspable. On the other hand, in the case of the search-for-grasp method, the target object's graspability is taken into account in the searching phase, so the two boxes are rearranged in a way that the target object becomes not only visible but also graspable as shown in the right of Figure 16. In summary, when the target object is occluded by multiple objects, the search-for-grasp method can be efficient in terms of the number of actions. To verify this, we design an additional simulation experiment named *3-object toy experiment* as described below.

**Object configuration.** We have created 200 scenarios in the pybullet simulation environment with two large cylindrical objects and one small cylindrical target object. The radius and height of the target cylinder is fixed, and those of two large cylinders are randomly selected big enough to occlude the target.

**Initial scene setting.** The position of the camera center and the positions of three cylindrical objects are on a straight line in the x-y plane so that the target object is occluded by the other two objects on the shelf. The position of the target object is fixed, while the $(x, y)$ coordinates of the other objects' positions are randomly selected on the straight line.

**Target detection.** Target detection method is the same with Appendix C.2.

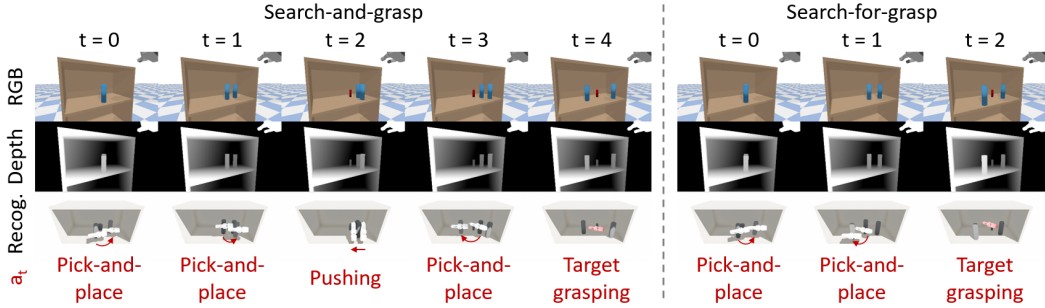

Figure 17: Example trajectories of simulation manipulation for R-search-and-grasp (*Left*) and R-search-for-grasp (*Right*). Each column shows the camera input and action selection at each time step. In the simulation, surrounding objects are blue and the target object is red.

Figure 17 shows the mechanical search results by search-and-grasp and search-for-grasp. In this subsection, we only use recognition-based (i.e., R-search-and-grasp and R-search-for-grasp). The search-and-grasp method succeeds in finding the target object in two pick-and-place actions as de-

sired, but when target object is found, it cannot be grasped due to collision with surrounding objects. So, this method makes the target object graspable by performing two additional actions as shown in the left of Figure 17. In the case of the search-for-grasp method, it also succeeds finding the target object in two pick-and-place actions, and at this time, the target object becomes graspable at the same time as shown in the right of Figure 17. This difference of the number of actions is due to the difference in whether the graspability of the found object is taken into account when searching for the object. Although the search-for-grasp method is slightly more computationally expensive than the search-and-grasp method, the search-for-grasp method can be efficient in terms of number of actions in these situations.

In order to quantitatively verify this fact, we measure (i) the average grasp-success rate and the average number of actions to (ii) find and (iii) grasp the target object for the two methods, and the results are shown in Table 3. We note that the minimum number of actions required to find and grasp is 2. The average number of

| METHOD | Succ. | Find | Grasp |
|---|---|---|---|
| R-Search-and-Grasp | 0.985 | **2.065** | 2.919 |
| R-Search-for-Grasp | **0.995** | 2.255 | **2.497** |

Table 3: Manipulation results for 3-object toy experiment.

actions to *find* is lower for search-and-grasp than for search-for-grasp since the search-and-grasp method only focuses on finding the target object. On the other hand, the average number of actions to *grasp* is lower for search-for-grasp than for search-and-grasp. Unlike the search-and-grasp methods, the search-for-grasp simultaneously search for a target object while increasing the probability of grasping the found target object. As a result, this fact has the effect of reducing the average number of actions. In conclusion, we verify that the search-for-grasp method can be more efficient than the search-and-grasp method in terms of the number of actions when the target object is occluded by multiple objects.

### D.2 Mechanical Search via Only Pushing or Only Pick-and-place

This experiment evaluates the performance of two cases where only one action type (e.g., pick-and-place or pushing) is allowed, demonstrating that both motions are essential for mechanical search tasks on shelves. We only use recognition-based method in this experiment.

**Object configuration.** We have created 75 scenarios for each number of surrounding objects in $\{4, 8\}$, so a total of 150 scenarios; for each scenario, a random selection (allowing duplicates) is made among the given objects.

**Action sampler.** Instead of action sampler described in B.5, if only push is allowed, the action is sampled only in $I_p$. If only pick-and-place is allowed, the action is sampled only in $I_g$.

**Initial scene setting** Initial scene setting is the same with Appendix C.2.

**Target detection.** Target detection method is the same with Appendix C.2.

Figure 18 shows the mechanical search results by our methods using only pick-and-place and only pushing. In the first example, using only pushing succeeds in finding and grasping the target object in one action as desired, but using only pick-and-place fails since there is no valid pick-and-place action. In the second example, using only pick-and-place succeeds in finding and grasping in one action, and using only pushing succeeds in finding the target object but there is no pushing action to move the blue cylinder to be removed since it becomes close to the wall of shelf. As such, there are cases where using only pick-and-place and only pushing show different trajectories.

To quantitatively investigate the roles of the pushing and pick-and-place, and the results are shown in Table 4. First, allowing only one type of action degrades the performance significantly as shown in the table. Specifically, using only pick-and-place highly degrades the performance of finding the target object compared to the case where both actions are used, and on the other hand, using only pushing highly degrades the performance of grasping the target object, even though the success rate in finding is higher than using only pick-and-place.

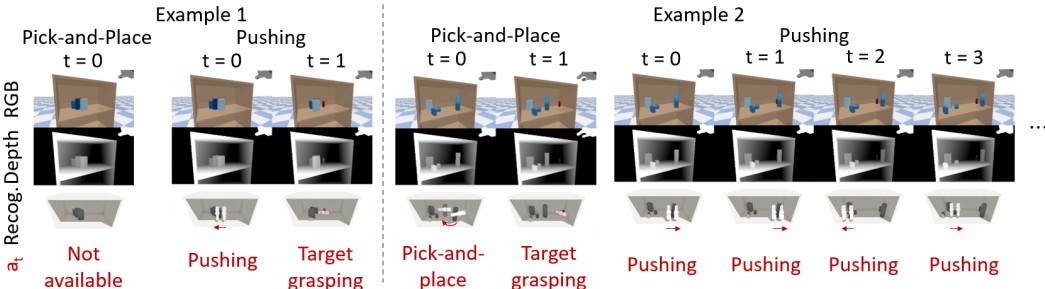

Figure 18: Example trajectories of simulation manipulation using only pick-and-place and only pushing. Each column shows the camera input and action selection at each time step. In the simulation, surrounding objects are blue and the target object is red. (*Left*) A scenario where only pick-and-place fails but only pushing succeeds. (*Right*) A scenario where only pick-and-place succeeds but only pushing fails.

| METHOD | ALLOWED ACTION | | The number of objects | | | |
|---|---|---|---|---|---|---|
| | | | 4 | | 8 | |
| | | | Find | Grasp | Find | Grasp |
| R-Search-and-Grasp | Both | Succ. | 0.973 | 0.787 | 0.893 | 0.613 |
| | | Steps | 2.164 | 2.559 | 3.269 | 3.957 |
| | Pick-and-Place | Succ. | 0.76 | 0.627 | 0.627 | 0.48 |
| | | Steps | 1.404 | 1.596 | 2.255 | 3.361 |
| | Push | Succ. | 0.907 | 0.573 | 0.813 | 0.507 |
| | | Steps | 2.559 | 3.628 | 3.098 | 4.763 |
| R-Search-for-Grasp | Both | Succ. | 0.96 | 0.813 | 0.92 | 0.6 |
| | | Steps | 2.139 | 2.77 | 3.101 | 3.889 |
| | Pick-and-Place | Succ. | 0.76 | 0.627 | 0.693 | 0.467 |
| | | Steps | 1.386 | 1.617 | 2.327 | 3.543 |
| | Push | Succ. | 0.92 | 0.613 | 0.773 | 0.56 |
| | | Steps | 2.609 | 3.826 | 3.034 | 4.5 |

Table 4: Simulation manipulation results for the one-type action experiments.

This result is because pick-and-place and pushing have different strengths and weaknesses. The pick-and-place is efficient in terms of the average number of actions because it can move the objects farther away, but the actions which can be performed are limited due to the fewer objects the robot can grasp in a cluttered scene. The pushing is better than pick-and-place for finding target objects because the robot can manipulate more objects than pick-and-place, but because the robot can't move the objects that far, it requires a higher number of actions because the can't move the objects that far. In summary, both pick-and-place and pushing are required to find the target and grasp the target object.

### D.3 Mechanical Search with Box Target Object

This experiment evaluates the performance of our methods for mechanical search tasks with a box target object. In this case, the target pose space should be $\mathcal{X} = \text{SE}(2)$, i.e. position $(x, y)$ of the target box on the shelf and its z-axis angle $\theta$ as orientation. We only use recognition-based method in this experiment.

**Object configuration.** We have created 40 scenarios for each number of surrounding objects in $\{4, 8\}$, so a total of 100 scenarios; for each scenario, a random selection (allowing duplicates) is made among the given objects. The target object is replaced with a box shape instead of a cylinder.

Other settings are following the section C.2.

Figure 19 shows the mechanical search results by search-for-grasp on the task of the box target object. The search-and-grasp method succeeds in finding the target object in three pick-and-place

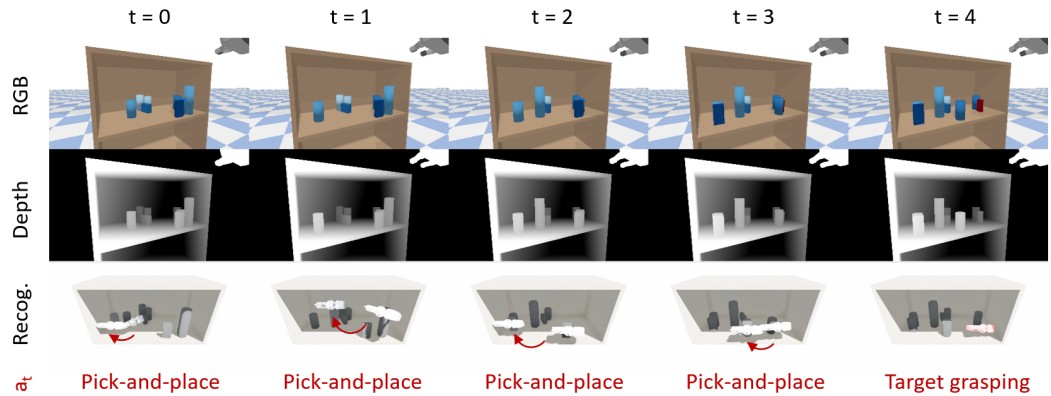

Figure 19: An example trajectory of simulation manipulation for R-search-for-grasp for the box-shaped target object. Each column shows the camera input and action selection at each time step. In the simulation, surrounding objects are blue and the target object is red.

| | | The number of objects | | | |
| | | 4 | | 8 | |
| **METHOD** | | Find | Grasp | Find | Grasp |
|---|---|---|---|---|---|
| R-Search-and-Grasp | Succ. | 0.925 | 0.775 | 0.825 | 0.525 |
| | Steps | 1.568 | 3.323 | 2.394 | 4.81 |
| R-Search-for-Grasp | Succ. | 0.95 | 0.7 | 0.85 | 0.6 |
| | Steps | 1.652 | 3.286 | 2.412 | 5.167 |

Table 5: Simulation manipulation results for the box target object

actions and succeeds in grasping the target object in additional one pick-and-place action. Table 5 shows the performance of our methods with the box target object. Compared to the cylinder target object case, the grasp success rate is lower when the number of objects is 4. This is because the approach direction of the grasping trajectory is limited to one in the case of the box as shown in the Figure 11; we note that the cylinder can be grasped in any direction due to rotational symmetry. However, when the number of objects are large (i.e., the number of objects is 8), the box target object case shows similar performance to the cylinder target object case. This is because the advantage of having various grasping approach directions of cylinders is lost in highly cluttered environment. In summary, we conclude that our algorithm also works on the target pose space of $SE(2)$ on the box target object experiment.

