# OpenReview forum: "Leveraging 3D Reconstruction for Mechanical Search on Cluttered Shelves"
_robot-learning.org/CoRL/2023/Conference — CoRL 2023 Poster_

### Official Review · Reviewer_bz1q · 2023-06-29

**Confidence:** 4
**Originality:** Good
**Technical Quality:** Very Good
**Clarity Of Presentation:** Good
**Impact:** 3

**Recommendation:**

Weak Accept: I recommend accepting the paper, but will not argue for my recommendation if the majority of other reviewers have a different opinion.

**Review:**

The paper leverages 3D reconstruction of the scene to estimate the existence function and graspability function, which enables the robot to find and grasp target objects in cluttered shelves without specialized tools. This is a practical and innovative approach that has not been explored extensively in previous research. The paper provides a well-structured framework for mechanical search and grasping, with specific algorithms for search-and-grasp and search-for-grasp methods. The formulation of the problem as a model-based optimal control problem is clear and logical. The experimental results in both simulation and real-world environments validate the effectiveness and real-world applicability of the proposed method. The comparison with cases where all object information is assumed to be known demonstrates that the proposed method performs similarly.

Since the abstract mentions that existing works primarily focus on specialized tools with suction grippers, it would be interesting to see a comparison between these works and the proposed algorithm based on 3D reconstruction. Providing this contextual background will help the reader understand the novelty and originality of the proposed method. The paper lacks sufficient implementation details for reproducing the study. It would be helpful to provide more information on the specific techniques and algorithms used in the 3D reconstruction, estimation of the existence function and graspability function, and model-based optimal control. The paper could benefit from more thorough evaluation and ablation studies. It would be valuable to compare the proposed method with widely-known baselines in the field and conduct additional experiments to further analyze the performance and limitations of the method.

**Quality Of The Limitations Section:**

Limitations are addressed clearly

**Questions For Rebuttal:**

1. Can the authors more details on the implementation of the 3D reconstruction and estimation of the existence function and graspability function? How was the pushing dynamics model and depth rendering utilized?
2. Have the authors compared the proposed method with existing baselines? It would be valuable to see how the proposed method performs compared to other approaches.

**Robotics Focus:**

Sufficient demonstration on hardware

**Summary Of Paper:**

This paper proposes a practical method for finding and grasping target objects on cluttered shelves using a standard robot gripper. The authors develop an updated mechanical search algorithm based on 3D reconstruction of the scene, which is used to estimate the existence function and graspability function of the target object. The authors formulate the problem as a model-based optimal control problem and provide algorithms for both search-and-grasp and search-for-grasp methods. Experimental results in both simulation and real-world environments demonstrate the effectiveness of the proposed method.

**Summary Of Recommendation:**

More details on baselines and implementations are needed.

---

> ### Author Response · Authors · 2023-08-12
> **Response to Reviewer bz1q**
>
> __Q1.__
> Since the abstract mentions that existing works primarily focus on specialized
> tools with suction grippers, it would be interesting to see a comparison between
> these works and the proposed algorithm based on 3D reconstruction. Providing this
> contextual background will help the reader understand the novelty and originality
> of the proposed method. It would be valuable to compare the proposed method with
> widely-known baselines in the field and conduct additional experiments to further
> analyze the performance and limitations of the method. Have the authors compared
> the proposed method with existing baselines? It would be valuable to see how
> the proposed method performs compared to other approaches.
>
> __A1.__
> The main difference from existing works is that the graspability of the target object is taken into consideration when performing mechanical search. The existing studies, including [19], use a custom long suction gripper specialized for mechanical search, so the graspability of the target object does not need to be considered separately. On the other hand, when using the standard robot gripper (e.g., parallel-jaw gripper), we need to find a trajectory that does not collide with the surrounding objects and the shelf to grasp the target object. To find a non-colliding trajectory, 3D reasoning of the current scene is inevitable for 6-DOF grasping, and accordingly, a 3D recognition model has been adopted in our method. The existing works only perform 2D reasoning for the scene since they do not have to consider graspability, and the fact that we perform a mechanical search task adopting 3D recognition is also different from other works. ``We have added these discussions to Appendix A (Related Works).``
>
> __Q2.__
> The paper lacks sufficient implementation details for reproducing the study.
> It would be helpful to provide more information on the specific techniques and
> algorithms used in the 3D reconstruction, estimation of the existence function
> and graspability function, and model-based optimal control. Can the authors more
> details on the implementation of the 3D reconstruction and estimation of the
> existence function and graspability function? How was the pushing dynamics model
> and depth rendering utilized?
>
> __A2.__
> Thank you for raising this issue. In the revised manuscript, we would like to mention that ``all implementation details of our method have now been added in more detail to Appendix B.`` Briefly, details for 3D reconstruction can be found in Appendix B.1 (Details for Object Shape Recognition). Details for the estimation of the existence function and graspability function can be found in Appendix B.2 (Details for Existence Function Estimate) and Appendix B.3 (Details for Graspability Function Estimate), respectively. Details for model-based optimal control and robot actions can be found in Appendix B.4 (Details for Action Sampling on Manipulation). We are now confident that the current version of the supplementary contains all the information needed to reproduce.
>
> __Q3.__
> The paper could benefit from more thorough evaluation and ablation studies.
>
> __A3.__
> We agree with the reviewer that several ablation studies can highlight the importance of our research. To address this issue, we conducted two ablation experiments and ``accordingly added Appendix D.1 (3-Object Toy Experiment) and Appendix D.2 (Mechanical Search via Only Pushing or Only Pick-and-place).``
>
> In Appendix D.1, we evaluate the performance of the two proposed methods, search-and-grasp and search-for-grasp, and especially highlight the advantages of the search-for-grasp method. Briefly, in this experiment, we confirmed that the search-for-grasp method outperforms the search-and-grasp method both qualitatively and quantitatively in terms of the number of robot actions when the target object is occluded by multiple objects.
>
> In Appendix D.2, we evaluate the performance of the two cases where only one type of action among pushing and pick-and-place is allowed, and highlight that both actions are essential for the mechanical search task on shelves. Briefly, we found that the pick-and-place action and the pushing action have different strengths and weaknesses in mechanical search tasks, and they complement each other when both actions are used.

---

### Official Review · Reviewer_e7zE · 2023-07-08

**Confidence:** 4
**Originality:** Fair
**Technical Quality:** Fair
**Clarity Of Presentation:** Excellent
**Impact:** 3

**Recommendation:**

Weak Reject: I recommend rejecting the paper, but will not argue for my recommendation if the majority of other reviewers have a different opinion.

**Review:**

This paper proposes a system for grasping occluded objects in shelves via rearrangement. I believe the paper has the following strengths:

- The paper’s methods and findings are presented in a clear and digestible manner.
- Physical experiments show the practicality of this system.
- Well-defined problem statement.
- Presents a principled approach to the problem of mechanical search for extracting a target object.

However, I have the following concerns about the proposed method and its fit for this conference.

- Given that 3D object recognition is the only learning-based component in this work, my feeling is that the paper’s emphasis should be more on the application of this method to mechanical search and what challenges & solutions are present in application to mechanical search. This is a good robotics paper, but I don’t know if it is suitable for CoRL given that the only learning is the application of an object detection model from prior work.
- The success rate demonstrated in the real world experiments is quite low for 5 and 6 objects. The failure cases include incorrect 3D recognition but do not elaborate further. Again, given that this is the only learning in the paper, it would be good to explain this further. In simulation, the success rate is much higher; why is this? Is object recognition in the real world harder for some reason?
- It is unclear to me what impact the superquadric-based object recognition method actually has. No ablations are provided to indicate whether simply segmenting objects and using that segmentation mask with some simple symmetry prior in 3d would give a sufficient shape to use for collision checking. Why does implicitly defining these objects’ shape via superquadrics help? Is there any indication that recognizing subtle variations in these objects’ shapes improves collision checking, etc. If not, isn’t the object recognition method overkill for the problem?

More minor points.

- Using the word “Verify” for experiments may signal some confirmation bias; better to use words like “evaluate” instead.
- Would be nice to have standard error or standard deviation in the table, or at least an explanation of what the distribution & variance for # of steps looks like.

**Quality Of The Limitations Section:**

Limitations are addressed clearly

**Questions For Rebuttal:**

- What are all the substantial differences between prior mechanical search work such as [19] and your proposed method? They feel quite similar. While hardware used is different, they also use a lookahead algorithm and similar pushing and pick & place actions. The main difference appears to be the actual grasping of the target object.
- In the video, certain rollouts are listed as failures after a few suboptimal actions. I am unsure of why the algorithm seems to be unable to recover from these situations.
- Please let me know if there are any factual inaccuracies in my understanding of your work :)

**Robotics Focus:**

Sufficient demonstration on hardware

**Summary Of Paper:**

This paper proposes a system for finding and grasping occluded objects in shelves without specialized hardware beyond a parallel jaw gripper. The authors propose an objective function that takes into account two functions, $f(x)$ and $g(x)$, representing existence and graspability. They then use shooting methods to minimize this objective over the course of a rollout. There are two versions of the method considered, one where graspability is considered as part of the objective, and one where it is considered as a separate step after visibility is achieved. The system employs 3d object recognition methods based on recognizing implicitly defined superquadrics, to recognize the geometry of objects within a scene at each step in the real world.

**Summary Of Recommendation:**

I think that the work has potential, but it would be good to see more novelty on the learning side for this problem beyond the object recognition, where it is unclear how useful the superquadric object detection method is. i.e. Which listed limitations could learning potentially help tackle?

---

> ### Author Response · Authors · 2023-08-12
> **Response to Reviewer e7zE**
>
> __Q3.__
> It is unclear to me what impact the superquadric-based object recognition
> method actually has. No ablations are provided to indicate whether simply
> segmenting objects and using that segmentation mask with some simple symmetry
> prior in 3d would give a sufficient shape to use for collision checking.
> Why does implicitly defining these objects’ shape via superquadrics help?
> Is there any indication that recognizing subtle variations in these objects’
> shapes improves collision checking, etc. If not, isn’t the object recognition
> method overkill for the problem?
>
> __A3.__
> We appreciate the reviewer’s question regarding “what impact the superquadric-based object recognition method actually has.” First, the superquadric-based object recognition model enables robust 3D object recognition in highly cluttered scenes from only partial observation. Since the back part of the shelf environment cannot be observed, objects can only be seen in a partial view. Additionally, objects occlude each other in a highly cluttered scene. Under this situation, it is difficult to determine the 3D shapes of individual objects, even if the symmetry prior of the object is assumed. The superquadric-based recognition model can quickly and robustly determine the shape of objects through a superquadric inductive bias on the output; we refer to Appendix B.1 for additional details on the superquadric-based recognition model.
>
> Second, and more importantly, the implicit representation of these superquadric objects makes it possible to quickly compute the existence function and graspability function used for optimal control. In detail, the computation of these functions consists of collision checking between the objects and the gripper or depth image rendering, which can be calculated rapidly given the implicit representations of the objects. To highlight the advantages of using these superquadrics, ``we have added detailed descriptions about the calculation of the existence function and graspability function in Appendix B.2 (Details for Existence Function Estimate) and Appendix B.3 (Details for Graspability Function Estimate), respectively.``
>
> __Q4.__
> Using the word “Verify” for experiments may signal some confirmation bias;
> better to use words like “evaluate” instead.
>
> __A4.__
> Thank you for this suggestion. `` We have modified some
> expressions as the reviewer commented.``
>
> __Q5.__
> What are all the substantial differences between prior mechanical search work
> such as [19] and your proposed method? They feel quite similar. While hardware
> used is different, they also use a lookahead algorithm and similar pushing and
> pick \& place actions. The main difference appears to be the actual grasping of
> the target object.
>
> __A5.__
> The main difference from existing works, such as [19], is that the graspability of the target object is taken into consideration when performing a mechanical search. Existing studies, including [19], use a custom long suction gripper specialized for mechanical search, so the graspability of the target object does not need to be considered separately. On the other hand, when using a standard robot gripper (e.g., a parallel-jaw gripper), we need to find a trajectory that does not collide with surrounding objects or the shelf to grasp the target object. To find a non-colliding trajectory, 3D reasoning of the current scene is inevitable for 6-DOF grasping; accordingly, a 3D recognition model has been adopted in our method. Existing works only perform 2D reasoning for the scene since they do not have to consider graspability. The fact that we perform a mechanical search task adopting 3D recognition also differentiates our approach from other works. ``We have added these discussions to Appendix A (Related Works).``
>
> __Q6.__
> In the video, certain rollouts are listed as failures after a few suboptimal
> actions. I am unsure of why the algorithm seems to be unable to recover from
> these situations.
>
> __A6.__
> Thank you for raising this issue. There are two types of failure cases. In the first case, an erroneous existence function is estimated because of inaccurate recognition, as answered in A2. As a result, the robot stops because there are no more actions to reduce the objective function. In the second case, neither pick-and-place nor pushing actions can be performed anymore (see the second failure case in the video). In other words, if all possible actions in our action space collide with other objects or the shelf, no more actions are performed. ``We have included these additional explanations about the failure cases in Section 4.2 (Real-world Experiments Results).``

---

> > ### Comment · Reviewer_e7zE · 2023-08-14
> > **Response to authors**
> >
> > Thank you for taking the feedback to heart. I know it's not easy to make this many updates to your paper on short notice, I truly appreciate the effort. Some notes below.
> >
> > - I forgot to include this in my original comments, but I feel that including related work in the body of the paper is very important, compared to putting it in the Appendix. I know the CoRL format is limiting, but I feel this is important. I know time is limited now and I don't expect this to be done right away, but for the camera-ready paper if accepted, it would be good.
> > - Q2: Thank you, I do feel that it would be better to potentially include case-by-case breakdowns to see what failure cases are most prominent though.
> > - Q4, Q5, Q6: Thank you for the responses, I found them satisfactory.
> > - Q1 & Q3: This is a point which I want to press on some more. I feel that it is difficult to make claims about the effectiveness of the superquadric-based object detection method unless baselines are provided. There don't seem to be ablations. This is something I feel quite strongly about, given this superquadric object detection method is claimed as a vital piece of the algorithm. It would be good to have data that backs this up in the form of ablations/baselines.

---

> > > ### Author Response · Authors · 2023-08-15
> > > **Additional Response to Reviewer e7zE**
> > >
> > > Thank you for your quick response. Below are the responses to the reviewer's additional comments.
> > > * As the reviewer suggests, we will add a simplified related works section after Section 1 (Introduction) in the camera-ready version.
> > > * We will include the details of the failure cases of the simulation or real-world experiments, e.g., what failure cases are most prominent, in the camera-ready version.
> > > * We want to highlight again that one of our contributions involves a novel combination of learning-based methods—for 3D recognition and pushing dynamics models—and the optimal control method MPC, leading to a general 3D mechanical search and 6-DOF grasping framework. Given the context, it appears beneficial to perform an ablation study on object recognition models including the superquadric object recognition method. For instance, there is a method that the reviewer suggested which utilizes symmetry prior to 3D shape, or there is a general 3D recognition approach involving 3D implicit functions [1]. We will attempt to include an ablation study with these methods in the camera-ready version. Thank you for your great suggestion.
> > >
> > > [1] Park, J. J., Florence, P., Straub, J., Newcombe, R., & Lovegrove, S. (2019). Deepsdf: Learning continuous signed distance functions for shape representation. In Proceedings of the IEEE/CVF conference on computer vision and pattern recognition (pp. 165-174).

---

> > > > ### Comment · Reviewer_e7zE · 2023-08-16
> > > > **Thank you**
> > > >
> > > > Thank you for the responses. I trust that you would do the experiments on various object recognition models. However, it's really difficult for me to anticipate that the results would demonstrate that superquadric-based object recognition does provide a meaningful performance boost without really seeing the numbers, especially because most of the objects shown are not too far from simple rectangular or cylindrical shapes, where graspability would probably not be affected too much if they were just approximated as such. Please let me know if I am misunderstanding anything.

---

> ### Author Response · Authors · 2023-08-12
> **Response to Reviewer e7zE**
>
> __Q1.__
> Given that 3D object recognition is the only learning-based component in this
> work, my feeling is that the paper’s emphasis should be more on the application
> of this method to mechanical search and what challenges \& solutions are present
> in application to mechanical search. This is a good robotics paper, but I don’t
> know if it is suitable for CoRL given that the only learning is the application
> of an object detection model from prior work.
>
> __A1.__
> Regarding the reviewer's concern about the suitability of our paper for CoRL, we would like to refer to the two recommended areas listed in the Call for Papers:
> * Combination of learning- and planning-based approaches in robotics;
> * Applications of robot learning in robot manipulation, navigation, locomotion, driving, flight, and other areas of robotics.
> Our work proposes a novel combination of learning-based methods—for 3D recognition and pushing dynamics models—and the optimal control method MPC, leading to a general 3D mechanical search and 6-DOF grasping framework. We believe our method introduces novel components from the above two aspects: a unique combination and a new application task.
>
> At first glance, our framework may seem to consist of two independent components for recognition and optimal control, and it might appear that any arbitrary 3D recognition models can be employed. However, the two steps are indeed highly correlated, and the proper combination is crucial for developing a practical system. Specifically, the two essential functionalities required in our framework are (i) the estimation of existence and graspability functions, which are the key (and novel) components in our methods, and (ii) dynamic models for those functions, to formulate a practically solvable MPC. Not all recognition models are suitable, and we believe it's a significant contribution that we have demonstrated that the superquadric recognition model is one of the ideal combinations for the above functionalities.
>
> In addition, to the best of our knowledge, we have formulated and addressed a novel application task in robot manipulation: the 3D mechanical search and 6-DOF grasping. While previous works have been limited to the 2D mechanical search and grasping with a suction gripper, we demonstrate that our combined learning and optimal control method can tackle the challenging 3D search and 6-DOF grasping task. We'd like to emphasize that the key to our success in this new challenging task is the right combination of learning and optimal control methods.
>
> __Q2.__
> The success rate demonstrated in the real world experiments is quite low for
> 5 and 6 objects. The failure cases include incorrect 3D recognition but do not
> elaborate further. Again, given that this is the only learning in the paper,
> it would be good to explain this further. In simulation, the success rate is
> much higher; why is this? Is object recognition in the real world harder for
> some reason?
>
> __A2.__
> Thank you for raising this point. As the reviewer commented, a major cause of the failure cases is inaccurate recognition. Some incorrect action decisions arise from erroneous existence map estimation. In the failure cases of our robot demo, there were instances where additional actions were not implemented even though they were possible. These are caused by erroneous existence map estimations stemming from inaccurate recognition results. For example, if the recognition model perceives an object as inaccurately large, the existence map may be underestimated; i.e., it is decided that $f(x)=0$, but the target object can exist at $x \in \mathrm{SE}(3)$. ``We have included these additional explanations about the failure cases in Section 4.2 (Real-world Experiments Results).``
>
> The recognition error arises because there are sim-to-real issues such as sensor noise, given that the recognition model was trained on simulation data. We refer to Appendix B.1 for additional details on the recognition model. Since we wanted the recognition results to be insensitive to lighting conditions or the issue of non-photorealistic training data, we only used depth images. If we incorporate RGB images in instance segmentation, we believe we can reduce the number of failure cases significantly. There are several available RGB-based instance segmentation models trained on real-world data [1, 2].
>
> [1] C. Xie, Y. Xiang, A. Mousavian, and D. Fox. Unseen object instance segmentation for robotic environments. IEEE Transactions on Robotics, 37(5):1343–1359, 2021.
>
> [2] A. Kirillov, E. Mintun, N. Ravi, H. Mao, C. Rolland, L. Gustafson, T. Xiao, S. Whitehead, A. C. Berg, W.-Y. Lo, et al. Segment anything. arXiv preprint arXiv:2304.02643, 2023.

---

### Official Review · Reviewer_QSGc · 2023-07-17

**Confidence:** 3
**Originality:** Very Good
**Technical Quality:** Good
**Clarity Of Presentation:** Very Good
**Impact:** 3

**Recommendation:**

Strong Accept: I recommend accepting the paper and will argue for my recommendation even if other reviewers hold a different opinion.

**Review:**

Originality: This is the first work I'm aware of that addresses mechanical search on cluttered shelves with standard grippers. The idea of using indicator functions to guide rearrangement actions is novel. Estimating these based on 3D recognition is also a new approach. The optimal control formulation using dynamics of learned indicator functions is unique.

Quality: The technical approach is sound, leveraging superquadric recognition, PyBullet physics, and model predictive control. The experiments evaluate performance in simulation and real-world settings. Limitations are properly acknowledged regarding gripper constraints and recognition errors.

Clarity: The paper is well-written and clearly conveys the key ideas, approach, and results. The problem is motivated early on, and technical sections provide sufficient details to understand the methodology.

Significance: This research tackles a difficult problem in robotic manipulation that arises in real-world conditions. The ability to mechanically search for and retrieve occluded objects in tight, cluttered spaces can enable many useful applications. The proposed methods advance the state of the art in using standard grippers rather than specialized tools.

Relevance: This work is relevant to the CoRL audience since it addresses a challenging robot learning problem involving mechanical search, interactive perception, and manipulation planning under uncertainty. The methods leverage learning for 3D recognition, dynamics modeling, and optimal control. Evaluation is extensive on both simulation and physical robots.

Limitations: As mentioned by the authors, the fixed grasping approach limits general applicability. Collision checking for the graspability function is also computationally expensive. Recognition errors can propagate. Larger-scale experiments would be useful to better characterize robustness. The superquadric representation also limits the object shapes. Only a quite limited variety of objects with limited complexity and poses is considered.

In summary, I believe this paper makes a good contribution in advancing mechanical search capabilities for retrieving occluded objects from cluttered shelves using standard grippers. The technical approach is novel, of good quality, and clearly conveyed.

**Quality Of The Limitations Section:**

Limitations are addressed clearly

**Questions For Rebuttal:**

Questions:

* Why use a learned model for pushing prediction rather than also simulating it like the pick-and-place actions?
* Have you experimented with more complex object shapes beyond basic geometries? How robust is the method to shape diversity?
* How is occlusion handled in the recognition model? Does it limit how cluttered the scene can be?
* What specific limitations did you observe in the real robot experiments? How can they be addressed?
* Does rearrangement strategy matter? Is your method optimized for minimally disturbing the scene?

Potential issues:

* Expand the object shapes used in experiments to better demonstrate generalizability.
* Provide ablation studies to isolate the impact of key algorithmic components.
* Benchmark on larger datasets to characterize robustness. How does performance degrade with scene complexity?
* Demonstrate benefits over ablations without the optimal control formulation.
* Compare to more baselines like end-effector picking without rearrangement.
* Discuss computational efficiency - how does it scale? Could optimizations be made?
* Validate more real-world factors like pose estimation noise, partial occlusions, novel objects.

**Robotics Focus:**

Sufficient demonstration on hardware

**Summary Of Paper:**

This paper tackles finding and grasping occluded objects on cluttered shelves using standard grippers. The key idea is representing target visibility and graspability as indicator functions and optimizing these via pushing and pick-place actions. 3D superquadric recognition enables estimating the functions from partial views. Model predictive control leverages the estimates to plan actions over a short horizon that reveal and enable grasping the target object. Experiments in simulation and real-world settings validate the proposed search-and-grasp and search-for-grasp methods can effectively retrieve objects in cluttered spaces comparable to oracle methods with full state knowledge. Key contributions are the visibility and graspability formulations, recognition-based estimation, and optimal control framework to enable mechanical search with standard grippers.

**Summary Of Recommendation:**

This paper makes technically strong contributions in advancing mechanical search capabilities using standard grippers rather than specialized tools. The novel formulation using visibility and graspability indicator functions provides a unique optimal control approach to sequential object rearrangement for retrieving occluded targets. Extensive experiments demonstrate effective performance on cluttered shelves in both simulation and real robot settings. While the object shapes are limited and impact potential incremental, the key ideas show promise in addressing a challenging problem. Overall quality and technical merit justify acceptance, though real-world applicability would benefit from more extensive validation and generalization beyond basic geometries.

---

> ### Author Response · Authors · 2023-08-12
> **Response to Reviewer QSGc**
>
> __Q4.__
> What specific limitations did you observe in the real robot experiments?
> How can they be addressed? Validate more real-world factors like pose estimation
> noise, partial occlusions, novel objects.
>
> __A4.__
> The major limitation of the real robot experiments is inaccurate recognition.
> First, there are some robot-object collisions shown in the real demo video.
> When performing a prehensile, pushing action, or target object retrieval,
> we check the collision between that robot action and the recognized objects; we refer to Appendix B.4 for additional details on the fast collision checking method. Therefore, the cause of some robot-object collisions is
> inaccuracies in the recognized objects. In our real robot experiment, this was
> not fatal (although the robot touched the objects a little), but it might later
> lead to objects falling over. Second, and more importantly, there are some wrong action decisions due to erroneous existence map estimation. For example, if the recognition model recognizes an object as inaccurately large, the existence map may be underestimated; that is, it is decided that $f(x)=0$, but the target object can exist at $x \in \mathrm{SE}(3)$. ``We have added these discussions and further clarified the limitations in Section 4.3 (Limitations and Future Directions).``
> The recognition error is caused by the fact that there are sim-to-real issues, such as sensor noise, since the recognition model was trained on simulation data. As we wanted the recognition results to be insensitive to lighting conditions or the issue of non-photorealistic training data, we used only depth images. If we use RGB images in conjunction with depth images for instance segmentation, we believe we can address these limitations in a meaningful way; several RGB-based instance segmentation models trained on real-world data are available [1, 2].
>
> [1] C. Xie, Y. Xiang, A. Mousavian, and D. Fox. Unseen object instance segmentation for robotic environments. IEEE Transactions on Robotics, 37(5):1343–1359, 2021.
>
> [2] A. Kirillov, E. Mintun, N. Ravi, H. Mao, C. Rolland, L. Gustafson, T. Xiao, S. Whitehead, A. C. Berg, W.-Y. Lo, et al. Segment anything. arXiv preprint arXiv:2304.02643, 2023.
>
> __Q5.__
> Does rearrangement strategy matter? Is your method optimized for minimally
> disturbing the scene?
>
> __A5.__
> Yes, the rearrangement strategy matters. In particular, in a setting where the pose of the camera is fixed, the robot has to find and grasp the target object by rearranging the surrounding objects inevitably. During rearrangement, we do not optimize separately to minimize disturbance of the scene.
>
> __Q6.__
> Provide ablation studies to isolate the impact of key algorithmic components.
> Demonstrate benefits over ablations without the optimal control formulation.
> Compare to more baselines like end-effector picking without rearrangement.
>
> __A6.__
> We believe that these comments pertain to the ablation studies of our method.
> We agree with the reviewer that such ablation studies can highlight
> the importance of our research. To address this concern, we conducted two ablation
> experiments and ``accordingly added Appendix D.1 (3-Object Toy
> Experiment) and Appendix D.2 (Mechanical Search via Only Pushing or Only
> Pick-and-place)``.
>
> In Appendix D.1, we compare the performance of the two proposed methods:
> search-and-grasp and search-for-grasp, and especially highlight the advantages
> of the search-for-grasp method. Briefly, in this experiment, we confirmed
> that the search-for-grasp method outperforms the search-and-grasp method
> both qualitatively and quantitatively in terms of the number of robot actions when the target object is occluded by multiple objects.
>
> In Appendix D.2, we evaluate the performance of the two scenarios where only one type of action, either pushing or pick-and-place, is allowed. We emphasize that both actions are essential for the mechanical search task on shelves. Briefly,
> we discovered that the pick-and-place action and the pushing action have distinct strengths and weaknesses in mechanical search tasks, and they complement each
> other when both actions are utilized.
>
> __Q7.__
> Discuss computational efficiency - how does it scale? Could optimizations be made?
>
> __A7.__
> ``In Appendices B.2 and B.3, we added a discussion about the computational cost of the existence function and the graspability function``, both of which are closely related to the calculation time of MPC. In particular, the existence function can run batch-wise for candidate target poses, allowing the existence-ability of these poses to be calculated in parallel."

---

> > ### Comment · Reviewer_QSGc · 2023-08-15
> > **Thank you**
> >
> > Thank you for taking the time to address all of my concerns and questions in detail and the addition of a great deal of important information to the appendix. If I understand correctly, there were no changes made to the original manuscript, but only to the appendix. If possible, I think it beneficial to include some of the discussion into the main text.

---

> > > ### Author Response · Authors · 2023-08-15
> > > **Additional Response to Reviewer QSGc**
> > >
> > > Thank you for your response. We have currently made changes in only Section 4.3 (Limitations and Future Directions) of the original main text by reflecting the reviewer's comments. The revised section addresses the discussions pointed out by the reviewer about the applicability to complex-shaped objects and limitations in real robot experiments. In the main text of the camera-ready version, we will attempt to include more discussions, especially, brief discussions about the ablation studies of our method in Appendix D.1 (3-Object Toy Experiment) and Appendix D.2 (Mechanical Search via Only Pushing or Only Pick-and-place). Thank you for your great suggestion.

---

> ### Author Response · Authors · 2023-08-12
> **Response to Reviewer QSGc**
>
> __Q1.__
> Why use a learned model for pushing prediction rather than also simulating
> it like the pick-and-place actions?
>
> __A1.__
> Thank you for raising this point. Before addressing this, we note that there was a revision regarding dynamics and scene prediction; when predicting the next scene after a pick-and-place action, we just transform the selected object rather than using the Pybullet simulator. In other words, for a pick-and-place action $(T_\text{grasp}, T_\text{place}) \in \mathrm{SE}(3) \times \mathrm{SE}(3)$,
> we predict the next scene by applying the transformation $T_\text{grasp}^{-1}
> T_\text{place}$ only to the selected object. `` We have added details on scene dynamics for pushing or pick-and-place action to Appendix B.4 (Details for Action Sampling on Manipulation). We regret that it was a confusing statement, and ``we have accordingly modified the description of the dynamics of the pick-and-place action in Section 3.3 (Approximate Dynamics Models).``
>
> Back to the point, the reason for using the learned model for pushing prediction is its fast inference speed. Since we used a sampling-based MPC, we should be able to quickly make predictions for a given action sequence. As the reviewer mentioned, there is also a way to simulate the pushing or pick-and-place action in the physical simulator to get the next scene, but this is too slow to be used for an optimal controller. Therefore, we decided to use a learned model for pushing prediction.
>
> __Q2.__
> Have you experimented with more complex object shapes beyond basic geometries?
> How robust is the method to shape diversity? Expand the object shapes used in
> experiments to better demonstrate generalizability.
>
> __A2.__
> Thank you for this suggestion. Since our method currently considers single superquadric-shaped objects, it is not trivial to apply it directly to more complex or non-convex shapes. One direction to extend our method is to exploit recent studies that attempt to represent complex objects using multiple superquadrics [1-4]. Even if each object is represented with several superquadrics, we can compute the existence function and the graspability function in the same manner. Extending our approach to handle multiple superquadric-shaped objects remains future work, and ``we have added commentary on the limitations and future directions of using superquadrics in Section 4.3 (Limitations and Future Directions).``
>
> [1] D. Paschalidou, A. O. Ulusoy, and A. Geiger. Superquadrics revisited: Learning 3d shape parsing beyond cuboids. In Proceedings of the IEEE/CVF Conference on Computer Vision and Pattern Recognition, pages 10344–10353, 2019
>
> [2] S. Kim, T. Ahn, Y. Lee, J. Kim, M. Y. Wang, and F. C. Park. Dsqnet: A deformable model-based supervised learning algorithm for grasping unknown occluded objects. IEEE Transactions on Automation Science and Engineering, 2022.
>
> [3] W. Liu, Y. Wu, S. Ruan, and G. S. Chirikjian. Robust and accurate superquadric recovery: a probabilistic approach. In Proceedings of the IEEE/CVF Conference on Computer Vision and Pattern Recognition, pages 2676–2685, 2022.
>
> [4] W. Liu, Y. Wu, S. Ruan, and G. S. Chirikjian. Marching-primitives: Shape abstraction from signed distance function. In Proceedings of the IEEE/CVF Conference on Computer Vision and Pattern Recognition, pages 8771–8780, 2023
>
> __Q3.__
> How is occlusion handled in the recognition model? Does it limit how cluttered
> the scene can be?
>
> __A3.__
> Thank you for raising this point. To address the reviewer's concern, ``we have added the implementation details of the object recognition model to Appendix B.1 (Details for Object Shape Recognition).`` Briefly, the recognition model is trained so that the predicted superquadric aligns well with the ground-truth full 3D shape, including the occluded parts. The loss function is designed to measure the difference between the prediction and the ground-truth object shapes. Moreover, since the recognition model takes as input (i) not only the point cloud of the object of interest but also (ii) the point cloud of the surrounding environment, such as surrounding objects, it aims to accurately predict the shape of the object even in a cluttered environment.

---

### Official Review · Reviewer_kape · 2023-07-17

**Confidence:** 4
**Originality:** Very Good
**Technical Quality:** Good
**Clarity Of Presentation:** Very Good
**Impact:** 4

**Recommendation:**

Weak Accept: I recommend accepting the paper, but will not argue for my recommendation if the majority of other reviewers have a different opinion.

**Review:**

Strengths:
1. The paper overall has a smooth logic flow and it is easy to follow.
2. The problem setting that the object of interest is not visible initially is hard yet interesting and practical. In addition, the shelf environments bring more challenges to the problem like limited workspace, limited relocation space and severe occlusion, etc. This problem happens a lot in real-world scenarios if the agent can only observe the environment through a fixed camera. Solving this task can truly benefit real robotics applications.
3. The setting where all objects except the object of interest are initially unknown is a good practice as that's what happened in the real world, the agent is more likely to have only the observation and its target.
4. Since the object is not visible at first, the design of the existence function is smart. Also, the formulation of the model-based optimal control problem and the modification based on it makes good sense. The usage of MPC also fits well in this problem.
5. The authors provide most of the design details of the proposed method.
6. The real-world demos are comprehensive enough to demonstrate both successes and failures.

Weaknesses:
1. The authors claim that one of the challenges in the shelf environment is that the agent can only make observations from the front direction which makes the object of interest not observable in many cases. This is actually because of the setup in this paper (fixed camera) but not the nature of the shelf environment. If the robot has an in-hand camera that can observe the environment from different poses(although they are all at the front of the environment), I am sure that the red cylinder in the environment shown in Figures 1 and 4 can be seen even without relocating any objects. If this is the case, the system can directly figure out the objects in collision with the grasping path and relocate them according. Thus, the authors should include this camera setting in the introduction section to better motivate the proposed solution.
2. Claiming the object pose $x \in SE(3)$ is not accurate. First, since the target object is always placed on the surface, it is $(x, y)$ out of $(x, y, z)$. Then in both the figures and the video, the target object is always a red cylinder placed in a straight-up pose. I don't see there are any rotations included (at least no row and pitch, I cannot tell about the yaw because it does not matter for this object), which means it is at best $(yaw)$ out of $(row, pitch, yaw)$.  In all, the only thing that is certainly included in the calculation is a 2D point $(x,y)$. Although the authors did mention it as $\mathbb{R}^2$ in the experiment section, it creates some inconsistency.
3. In the search-for-grasp method, there are two cases where $f_t(x) = 1$, 1. the target object is observed 2. the target object is still hidden but occluded by others. Since the overall objective of the problem is to retrieve the target object, whenever the target object is observed, the agent should directly aim to clear the path and get it out of the shelf. In the current method, since it just uses $f_t(x)$, the agent may still make efforts to perform other actions instead of directly clearing the path to the target object. Having a design that distinguishes between these two cases can benefit the method (minimizing $J_t$ until the objects show up, there is no need to keep doing it at this point, like the search-and-grasp method).
4. There are some robot-object collisions shown in the real demo video. Once some objects fell due to collision, it can cause problems that can potentially prevent the robot from achieving the objective.
5. The paper does not have a novel core learning-based module. The pushing dynamics model and the object recognition model are adapted from other works (not sure if there are any modifications introduced). The depth segmentation network is based on the DGCNN, not sure if there are any modifications introduced. Learning is of great usage in the pipeline but not the one of the main contributions.

**Quality Of The Limitations Section:**

Additional details required

**Questions For Rebuttal:**

1. It's better for the authors to revise the structure of the paper by having a proper related work section.
2. The authors are encouraged to add the fixed camera setting in the introduction when talking about the challenges (the target object is not visible) that shelf environments introduced.
3. Please revise the candidate pose issue mentioned in the weaknesses section or provide additional details/experiments to support the $SE(3)$ claim.
4. It will be beneficial to give some explanation about why the object-to-robot collisions happen in the real-world demo. Do they come from the sensor noise or the inaccurate object models or some other facts in the motion planning method you used? The reasonable can be added to the limitation section.
5. How does the agent choose between pushing action and picking-and-place action during the MPC process? Do they have the same chance of being selected? Additional info is needed for the readers to fully understand the pipeline.
6. In line 216, I think the authors want to say Appendix Part C instead of B. Also, Appendix B.1 is empty. Consider removing or adding related content.
7. The real-world instance segmentation in cluttered shelf environments is challenging because of the merge of edges between objects as well as the poor lighting conditions. The author can provide additional info other than just saying it is a DGCNN if there are modifications applied. Although this is not the contribution of this work, but it is the key to making the real-world experiment work.
8. In line 312, what does it mean when the authors mention the gripper holds the object in a straight line? Does it mean the gripper will only try to approach the object from the front but not from the side?
9. The work itself is convincing. But the authors are encouraged to provide justifications for why this work is suitable for CoRL as the novelty in the robot learning part is limited.

**Robotics Focus:**

Sufficient demonstration on hardware

**Summary Of Paper:**

This paper proposes a mechanical search method in cluttered shelf environments where the object of interest is initially not visible. The system moves other objects in the scene through prehensile and pushing gripper actions to find and clear the path to the target object. The authors formulate the task into a model-based optimal control problem and come up with the existence function/graspability function, which works jointly as the cost function of the solver. MPC is used to find the best local robot action that fulfills the objective. The authors conduct experiments in both the simulation and the real-world robotics system. Promising results are presented to demonstrate the effectiveness of the proposed method.

**Summary Of Recommendation:**

The work is convincing and provides new ideas to solve the object grasping tasks in cluttered shelf environments by formulating it as a model-based optimal control problem. The proposed method is technically sound. Results from the simulation and the real-world robot demonstrate the effectiveness of the method. This work has enough contributions to the TAMP area in the robotics field. However, I think the novelty in the robot learning part is limited or not addressed enough in the current version of the paper. Thus, to the best of my knowledge, my recommendation is weak accept considering its contribution to the robotics field.

---

> ### Author Response · Authors · 2023-08-12
> **Response to Reviewer kape**
>
> __Q6.__
> It's better for the authors to revise the structure of the paper by having a
> proper related work section.
>
> __A6.__
> Thank you for your suggestion. Unfortunately, due to the page limit of the CoRL format, it is difficult to include it in the main text. Nevertheless, to the best of our knowledge, we have tried to find all of the studies related to our method and have compared them to our method in Appendix A (Related Works).
>
> __Q7.__
> How does the agent choose between pushing action and picking-and-place action
> during the MPC process? Do they have the same chance of being selected?
> Additional info is needed for the readers to fully understand the pipeline.
>
> __A7.__
> Thank you for the question. ``We have added details on action sampling methods (including pushing actions and pick-and-place actions) during the MPC process to Appendix B.4 (Details for Action Sampling on Manipulation).``
> Briefly, we first check which objects can be grasped (for the pick-and-place action) or pushed. We then sample from the graspable and pushable objects. After that, we select one possible action for each sampled object. During the MPC, multiple action sequences are sampled in a similar manner."
>
> __Q8.__
> In line 216, I think the authors want to say Appendix Part C instead of B.
> Also, Appendix B.1 is empty. Consider removing or adding related content.
>
> __A8.__
> Thank you for pointing it out. In the revised paper, ``we have modified the main text to direct readers to the appropriate appendix.``
>
> __Q9.__
> The real-world instance segmentation in cluttered shelf environments is
> challenging because of the merge of edges between objects as well as the poor
> lighting conditions. The author can provide additional info other than just
> saying it is a DGCNN if there are modifications applied. Although this is not
> the contribution of this work, but it is the key to making the real-world
> experiment work.
>
> __A9.__
> Thank you for raising this point. ``We have added the implementation details of the instance segmentation to Appendix B.1 (Details for Object Shape Recognition).`` We trained the instance segmentation model (i.e., DGCNN-based model) and the recognition model using simulation data, and applied the model directly to the real-world experiment. Since we wanted the recognition results to be insensitive to lighting conditions or the issue of non-photorealistic training data, we only used depth images. We hope that these additional contents are helpful.
>
> __Q10.__
> In line 312, what does it mean when the authors mention the gripper holds the
> object in a straight line? Does it mean the gripper will only try to approach
> the object from the front but not from the side?
>
> __A10.__
> We mean that the gripper approaches the object in a straight-line trajectory when trying to grasp it. We are aware that there was little description of grasping trajectories in the current version of the manuscript, so ``we have added detailed descriptions about grasping trajectories in Appendix B.3 (Details for Graspability Function Estimate).`` Returning to the limitation section, we check the graspability of an object using only a finite number of grasping trajectories. Therefore, there can be some cases where the object is decided to be not graspable even though there is actually a trajectory that can grasp the object. To clarify, ``we have modified the first limitation on graspability in Section 4.3 (Limitations and Future Directions).``

---

> > ### Comment · Reviewer_kape · 2023-08-15
> > **Response to authors**
> >
> > Thank you for addressing all the concerns in a prompt manner. I am satisfied with the responses and they make the modules introduced clearer to the readers. I don't have any additional comments at this point.

---

> > > ### Author Response · Authors · 2023-08-16
> > > **Additional Response to Reviewer kape**
> > >
> > > Thank you for your response. With the comments from the reviewers, we feel that the paper has been considerably improved. Additionally, concerning Q6, we will attempt to add a simplified related works section after Section 1 (Introduction) in the camera-ready version.

---

> ### Author Response · Authors · 2023-08-12
> **Response to Reviewer kape**
>
> __Q4.__
> There are some robot-object collisions shown in the real demo video. Once
> some objects fell due to collision, it can cause problems that can potentially
> prevent the robot from achieving the objective. It will be beneficial to give
> some explanation about why the object-to-robot collisions happen in the
> real-world demo. Do they come from the sensor noise or the inaccurate object
> models or some other facts in the motion planning method you used? The
> reasonable can be added to the limitation section.
>
> __A4.__
> Thank you for raising this point. Before addressing this, we would like
> to mention that the ``implementation details of our method have been
> added in more detail to Appendix B.`` When performing a prehensile, pushing
> action, or target object retrieval, we check the collision between the robot
> action and the recognized objects. For example, when executing a pick-and-place
> action, we first generate candidate grasping trajectories to grasp the selected
> object and then check whether there is a collision between the recognized objects
> and the afterimage of the robot trajectory when the robot follows its grasping
> trajectory.
>
> We note that the reason for some robot-object collisions is that the recognized
> object is inaccurate. This recognition error is caused by (i) the recognition
> model dealing with novel and highly-occluded objects that were not experienced
> during training and (ii) sim-to-real issues such as sensor noise,
> since the recognition model was trained on simulation data. This is one of
> our method's limitations as the reviewer noted. ``We have added
> these discussions about these limitations to Section 4.3 (Limitations and
> Future Directions).``
>
> __Q5.__
> The paper does not have a novel core learning-based module. The pushing dynamics
> model and the object recognition model are adapted from other works (not sure
> if there are any modifications introduced). The depth segmentation network
> is based on the DGCNN, not sure if there are any modifications introduced.
> Learning is of great usage in the pipeline but not the one of the main
> contributions. The authors are encouraged to provide justifications for why
> this work is suitable for CoRL as the novelty in the robot learning part is
> limited.
>
> __A5.__
> Thank you for raising this point. ``We have added the implementation
> details of the object recognition model to Appendix B.1 (Details for Object
> Shape Recognition).`` As the reviewer understood, the pushing dynamics model
> and the object recognition model are adapted from existing works; we have clarified these points in the current version of the supplementary material.
>
> Regarding the reviewer's concern about the suitability of our paper for CoRL,
> we would like to refer to the two recommended areas listed in the Call for
> Papers:
> * Combination of learning- and planning-based approaches in robotics;
> * Applications of robot learning in robot manipulation, navigation,
> locomotion, driving, flight, and other areas of robotics.
> Our work proposes a novel combination of learning-based methods -- for 3D
> recognition and pushing dynamics models -- and the optimal control method MPC,
> leading to a general 3D mechanical search and 6-DOF grasping framework.
> We believe our method incorporates novel components in both aspects:
> a unique combination and a novel application task.
>
> At first glance, our framework may seem to consist of two independent
> components for recognition and optimal control, and it might appear that arbitrary 3D recognition
> models can be employed. However, these two steps are indeed highly correlated,
> and the right combination is crucial for developing a practical system. Specifically,
> the two essential functionalities required in our framework are (i) the
> estimation of the existence and graspability functions -- which are key
> (and novel) components of our methods -- and (ii) dynamics models for those
> functions, to formulate a practically solvable MPC. Not all recognition models
> are suitable, and we believe it is an important contribution to demonstrate
> that the superquadric recognition model is one of the best
> combinations for these functionalities.
>
> Moreover, to the best of our knowledge, we have formulated and solved
> a novel application task in robot manipulation: the 3D mechanical search
> and 6-DOF grasping. While previous works have been limited to 2D mechanical
> search and grasping with a suction gripper, we demonstrate that our combined learning
> and optimal control method can address the challenging 3D search and 6-DOF
> grasping task. We want to emphasize that the key to our success in this
> new challenge is the precise combination of learning and optimal control
> methods.

---

> ### Author Response · Authors · 2023-08-12
> **Response to Reviewer kape**
>
> __Q2.__
> Claiming the object pose $x \in SE(3)$ is not accurate. First, since the target
> object is always placed on the surface, it is $(x, y)$ out of $(x, y, z)$. Then
> in both the figures and the video, the target object is always a red cylinder
> placed in a straight-up pose. I don't see there are any rotations included (at
> least no row and pitch, I cannot tell about the yaw because it does not matter
> for this object), which means it is at best $(yaw)$ out of $(row, pitch, yaw)$.
> In all, the only thing that is certainly included in the calculation is a 2D
> point $(x, y)$. Although the authors did mention it as $\mathbb{R}^2$ in the
> experiment section, it creates some inconsistency. Please revise the candidate
> pose issue mentioned in the weaknesses section or provide additional
> details/experiments to support the $SE(3)$ claim.
>
> __A2.__
> Thank you for pointing out this issue. We have attempted to describe the problem definition in general, considering cases where the target object can be tilted (i.e., non-zero $row$ and $pitch$ of the object orientation) or stacked on top of other objects (i.e., non-zero $z$ coordinate of the bottom of the objects). For this general description, the candidate poses are described as a subset of $SE(3)$. Currently, we provide experimental results with cylindrical target objects placed in a straight-up pose and not stacked (the target pose space is $\mathbb{R}^{2}$). Therefore, it is not sufficient to support the $SE(3)$ claim as the reviewer notes. To address this issue, `` we have slightly modified the explanation about assumptions and the target pose space in Section 2 (A General Framework for Mechanical Search and Grasping) and added Appendix D.3 (Mechanical Search with Box Target Objects).``
>
> In Section 2, we first add the assumption that the target object is placed in an upright pose (i.e., not tilted) and does not stack on top of other surrounding objects. Under this assumption, the space of the candidate target object pose is narrowed to $SE(2)$ (e.g., $(x, y, yaw)$) instead of $SE(3)$. We believe this modification will avoid reviewer confusion. However, we currently do not confirm that our method works in the target pose space of $SE(2)$. In Appendix D.3, we provide additional experimental results with box target objects to verify this. Unlike cylindrical objects, box objects do not have rotational symmetry, so it is important to consider orientation as well. Therefore, the target pose space is set to a subset of $SE(2)$ in this experiment. We have confirmed that our method also works to some extent on box target objects; we refer to Appendix D.3 for additional details and experimental results. We hope that these additions have further clarified the target object space.
>
> __Q3.__
> In the search-for-grasp method, there are two cases where $f_t(x)=1$, 1. the target object is observed 2. the target object is still hidden but
> occluded by others. Since the overall objective of the problem is to
> retrieve the target object, whenever the target object is observed, the
> agent should directly aim to clear the path and get it out of the shelf.
> In the current method, since it just uses $f_t(x)$, the agent may still
> make efforts to perform other actions instead of directly clearing the path
> to the target object. Having a design that distinguishes between these two
> cases can benefit the method (minimizing $J_t$ until the objects show up,
> there is no need to keep doing it at this point, like the search-and-grasp
> method).
>
> __A3.__
> If I understood the reviewer's comment correctly, the reviewer's concern is
> that when the target object is found at the pose $x^*$, there exist other
> $x$'s whose existence function $f(x)$ has a value of 1 (i.e., $f(x)=1, x
> \neq x^*$), so our optimal controller performs other actions instead of
> directly making the target object graspable. First, although it may already
> be apparent to the reviewer, we want to clarify that, when the target
> object is detected, the existence function $f$ becomes 1 only at one
> $x^*$ and $0$ for all other $x$'s. In this case, the objective function of
> the search-for-grasp method, $\sum_x f_T(x) + \alpha f(x)(1 - g(x))$,
> becomes just $1 + \alpha - \alpha g(x^*)$, where $g$ is the graspability
> function. We note that the existence function remains with a value of 1
> only at $x^*$ even if the target object is re-occluded due to additional
> action. In conclusion, after the target object is observed, our optimal
> controller only focuses on making the target object at $x^*$ graspable.
> We refer to Section 3.2 for the relation between whether the target object
> is visible and the existence function.

---

> ### Author Response · Authors · 2023-08-12
> **Response to Reviewer kape**
>
> __Q1.__
> The authors claim that one of the challenges in the shelf environment is
> that the agent can only make observations from the front direction which
> makes the object of interest not observable in many cases. This is actually
> because of the setup in this paper (fixed camera) but not the nature of
> the shelf environment. If the robot has an in-hand camera that can observe
> the environment from different poses(although they are all at the front of
> the environment), I am sure that the red cylinder in the environment shown
> in Figures 1 and 4 can be seen even without relocating any objects. If this
> is the case, the system can directly figure out the objects in collision
> with the grasping path and relocate them according. Thus, the authors should
> include this camera setting in the introduction section to better motivate
> the proposed solution.
>
> __A1.__
> Thank you for raising this issue. We agree with the reviewer's comment that the fixed camera setup can also be one of the factors that make the mechanical searching problem challenging. As the reviewer noted, the in-hand camera may allow us to find the target object using a new type of additional action (e.g., the active perception method described in Appendix A.1) in some cases. Although we cover the cases even where the target object cannot be found by only controlling the pose of the camera, it would be good to mention a fixed camera setting for this reason. ``We have accordingly added the fixed camera setting in Section 1 (Introduction) to better describe the challenges of the proposed problem and to motivate the proposed solution.``

---

### Decision · Program_Chairs · 2023-08-30

**Decision:**

Accept (Poster)

**Comment:**

Reviewers felt that the paper tackles an interesting and challenging problem, with a reasonable and well-explained solution.  The biggest concern was that it is unclear whether Superquadrics are important for this framework and whether there are any quantifiable advantages over standard methods for object recognition.